

# Representation learning with unconditional denoising diffusion models for dynamical systems

Tobias Sebastian Finn[1], Lucas Disson[1], Alban Farchi[1], Marc Bocquet[1], and Charlotte Durand[1]

[1]CEREA, École des Ponts and EDF R&D, Île-de-France, France

**Correspondence:** Tobias Sebastian Finn (tobias.finn@enpc.fr)

**Abstract.** We propose denoising diffusion models for data-driven representation learning of dynamical systems. In this type of generative deep learning, a neural network is trained to denoise and reverse a diffusion process, where Gaussian noise is added to states from the attractor of a dynamical system. Iteratively applied, the neural network can then map samples from isotropic Gaussian noise to the state distribution. We showcase the potential of such neural networks in experiments with the Lorenz 63 system. Trained for state generation, the neural network can produce samples, almost indistinguishable from those on the attractor. The model has thereby learned an internal representation of the system, applicable on different tasks than state generation. As a first task, we fine-tune the pre-trained neural network for surrogate modelling by retraining its last layer and keeping the remaining network as a fixed feature extractor. In these low-dimensional settings, such fine-tuned models perform similarly to deep neural networks trained from scratch. As a second task, we apply the pre-trained model to generate an ensemble out of a deterministic run. Diffusing the run, and then iteratively applying the neural network, conditions the state generation, which allows us to sample from the attractor in the run's neighbouring region. To control the resulting ensemble spread and Gaussianity, we tune the diffusion time and, thus, the sampled portion of the attractor. While easier to tune, this proposed ensemble sampler can outperform tuned static covariances in ensemble optimal interpolation. Therefore, these two applications show that denoising diffusion models are a promising way towards representation learning for dynamical systems.

## 1 Introduction

The ultimate goal of generative modelling is to generate samples from the distribution that has generated given training samples. Given this goal, we can train deep neural networks (NNs) for unconditional generation of states from the attractor of a dynamical system. Their further use beyond generating states remains ambiguous. Here, we reason that they learn an internal representation of the attractor. Instantiating denoising diffusion models (DDMs), we use the learned representation in downstream tasks, namely surrogate modelling and ensemble generation.

DDMs are trained to imitate the process of generating samples from the attractor of a dynamical system (Song et al., 2021), as depicted in Fig. 1a. During training, the available state samples are diffused by a pre-defined Gaussian diffusion kernel, and the NN is trained to denoise the diffused samples (Sohl-Dickstein et al., 2015; Ho et al., 2020). After training, we can iteratively apply the so-trained NN to map samples form a normal distribution to samples like drawn from the attractor. This



## (a) Unconditional denoising diffusion model

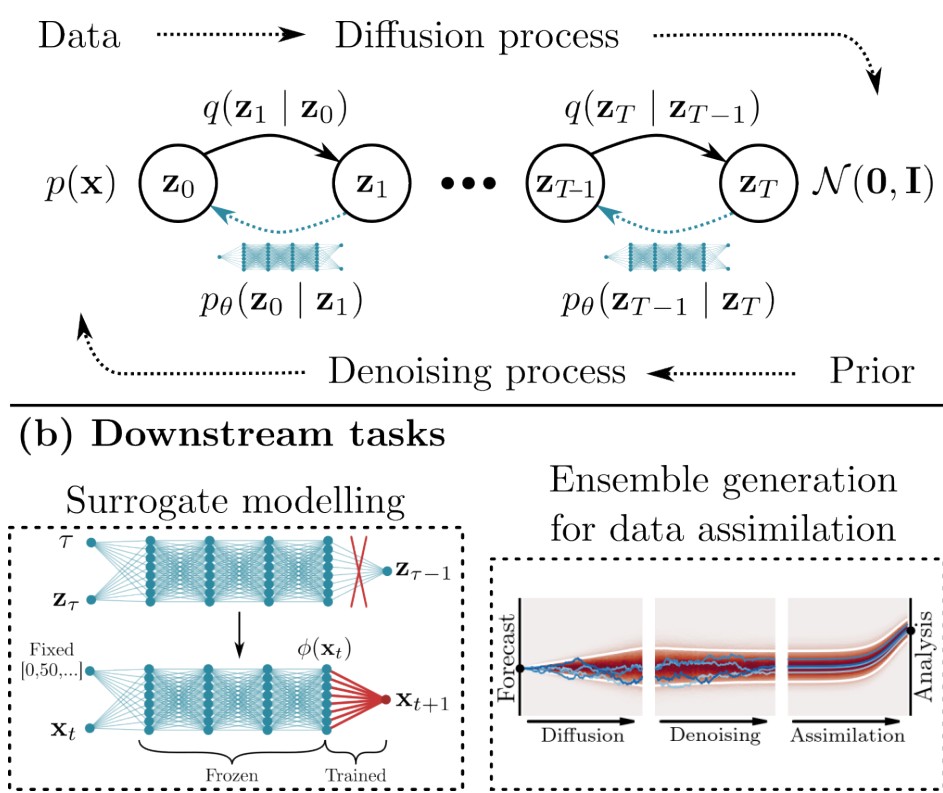

**Figure 1.** (a) We pretrain a neural network as a denoising diffusion model to generate states from the Lorenz 1963 system: during the diffusion process, Gaussian noise is increasingly added to the states until all information contained in the samples is completely destroyed. A neural network is trained to revert the process and to denoise diffused state samples. This denoising neural network can then generate new states from samples of a normal distribution. (b) We also use the denoising diffusion models for downstream tasks, different from state generation. We fine-tune the denoising network for surrogate modelling, and we apply the denoising diffusion model to generate an ensemble out of a deterministic run, e.g., for data assimilation.

generation process is akin to integrating a stochastic differential equation in (pseudo) time, where the NN defines the integrated dynamical system.

Using this correspondence between dynamical system and DDMs, we can replace the drift in the diffusion process by an integration of a real dynamical system (Holzschuh et al., 2023). The denoising process can then invert and integrate the system backward in physical time. Furthermore, DDMs can emulate fluid flows as simulated by computational fluid dynamics (Yang
and Sommer, 2023) and estimate a spatial-temporal prediction of such flows (Cachay et al., 2023). DDMs are additionally connected to the Schrödinger Bridge and can be designed to map between arbitrary probability distributions (Bortoli et al., 2021; Chen et al., 2023).



Generative modelling is a special case of self-supervised learning with the task of generating new state samples. Typically used for pre-training and representation learning, one of the promises of unsupervised and self-supervised learning is to learn deep NNs from large, heterogeneous datasets without the explicit need of supervision. Such methods allow us the use of NNs with millions of parameters for specific geoscientific problems (Hoffmann and Lessig, 2023; Nguyen et al., 2023), where often not enough labelled data is available to train deep NNs from scratch. Since training deep generative models remains difficult yet, generative training is rarely used for pre-training and representation learning of high-dimensional systems. DDMs offer a way for stable generative training and can generate high-quality samples (Dhariwal and Nichol, 2021; Nichol and Dhariwal, 2021). Hence, they have the potential to pave the way towards representation learning with generative models for high-dimensional systems.

DDMs are directly linked to denoising autoencoders (Vincent et al., 2008, 2010). These autoencoders train a NN to predict cleaned state samples out of noised ones; the NN must learn relevant features about the state distribution itself. These features are then useable in tasks different from denoising (Alain and Bengio, 2014; Bengio et al., 2013). The idea to reconstruct from corrupted data is further the leading paradigm in pre-training large language models (Radford et al., 2018; Devlin et al., 2019; Dong et al., 2019) and, recently, also used for high-dimensional image data with masked autoencoders (He et al., 2021). Based on these ideas, DDMs that are trained to generate images can extract useful features for downstream tasks (e.g., Baranchuk et al., 2022; Zhang et al., 2022; Xiang et al., 2023).

In our first downstream task, we follow along these lines and apply the denoising NN as feature extractor for surrogate modelling, as schematically shown in Fig. 1b. Initially pre-trained to generate states, we fine-tune the NN by replacing its last layer by a linear regression or a shallow NN. This way, we achieve a similar performance to that of deep neural networks trained from scratch.

In our second downstream task, we apply DDMs to generate state ensembles. Ensemble forecasting is one of the cornerstones for the recent advances in numerical weather prediction and data assimilation (Bauer et al., 2015), yet it is much more expensive than running a deterministic forecast. Ensemble optimal interpolation approaches (Evensen, 2003; Oke et al., 2002) lower the computational costs by applying climatological ensembles to assimilate observations into a deterministic forecast. The ensemble can be either directly drawn from the climatology or constructed by analogous methods (Lguensat et al., 2017; Tandeo et al., 2023). Alternatively, we can generate the ensemble members from the latent space of a variational autoencoder (VAE, Grooms, 2021; Yang and Grooms, 2021; Grooms et al., 2023).

In fact, DDMs are a type of (hierarchical) VAE with an analytically known encoding distribution (Kingma et al., 2021; Luo, 2022). Thus, the latent space of DDMs is similar to the latent space of VAEs, and image data can be interpolated in and reconstructed from this latent space (Song et al., 2020a). Mapping through this latent space by diffusing and denoising, we can perform image-to-image translation without the need of paired data (Meng et al., 2022).

Instead of image-to-image translation, we can also generate an ensemble out of a deterministic run with a DDM. We partially diffuse the run for a pre-defined amount of time. The mapping from state sample to diffused sample is inherently stochastic, and we can generate many diffused ensemble members. Afterwards, we iteratively apply the denoising NN to map the diffused ensemble back into state space. Varying the amount of time, we have control over the uncertainty in the generated ensemble. We





apply this ensemble sampling method in ensemble optimal interpolation to update a deterministic forecast. We demonstrate that so-generated ensemble members can outperform members drawn from tuned climatological covariances for ensemble optimal
interpolation.

In concurrent work from Rozet and Louppe (2023), the state generation with DDMs is guided towards observations. However, their prior distribution is defined by the climatology of their DDM, unconditional from any forecast run. To condition the denoising diffusion model on forecasts, Bao et al. (2023) retrain the network for each state update step. This retraining increases the computational costs of the data assimilation scheme and needs many ensemble samples. By contrast, we condition
the ensemble generation on a deterministic run using partial diffusion without the need to retrain the model.

We elucidate on the theory of denoising diffusion models in Sect. 2, where we additionally elaborate on different options for sampling and parameterizations of the NN output. In Sect. 3, we introduce our two methods to use the learned internal representation. There, we illustrate how the denoising NN is applied as feature extractor and how DDMs can generate an ensemble out of a deterministic run. Our experiments with the Lorenz 1963 system are described and analyzed in Sect. 4. We
summarize this work and discuss its broader impacts in Sect. 5, and we briefly conclude in Sect. 6.

## 2    Denoising diffusion models

Our goal is to generate state samples $\boldsymbol{x}$ as drawn from the attractor of a dynamical system. The distribution of states on the attractor is described by $p_{\text{data}}(\boldsymbol{x})$. This state distribution is unknown, and, instead, we rely on $k$ existing samples $\boldsymbol{x}_{1:k}$.

To generate state samples, we train deep neural networks (NNs) as denoising diffusion models (DDMs). Their general idea
for training is to progressively add noise to the training samples in a Gaussian diffusion process. This introduces a pseudo-time and results into noised samples $\boldsymbol{z}_\tau$ at a given step $\tau$. The NN $f_{\boldsymbol{\theta}}(\boldsymbol{z}_\tau, \tau)$ with its parameters $\boldsymbol{\theta}$ is trained to reverse the diffusion process and to denoise the samples for a single step. One denoising step can be described as follows,

$$p_{\boldsymbol{\theta}}(\boldsymbol{z}_{\tau-1} \mid \boldsymbol{z}_\tau) = \mathcal{N}(\mu_{\boldsymbol{\theta}}(\boldsymbol{z}_\tau, \tau), \Sigma_\tau), \tag{1}$$

here defined as a Gaussian distribution with mean $\mu_{\boldsymbol{\theta}}(\boldsymbol{z}_\tau, \tau)$ as a function of the NN output and $\Sigma_\tau$ as pseudo-time-dependent
covariance matrix. We will further discuss the parameterization of the NN output in Sect. 2.4.

After the NN is trained, we can start to sample from a known prior distribution $p(\boldsymbol{z}_T)$ and iteratively apply the NN for $T$ steps to denoise these samples towards the state space. This iterative sampling scheme results into the trajectory $\boldsymbol{z}_{0:T}$ with its joint distribution,

$$p_{\boldsymbol{\theta}}(\boldsymbol{z}_{0:T}) = p(\boldsymbol{z}_T) \prod_{\tau=1}^{T} p_{\boldsymbol{\theta}}(\boldsymbol{z}_{\tau-1} \mid \boldsymbol{z}_\tau). \tag{2}$$

In the following, we define the algorithm step-by-step by briefly explaining the diffusion process in Sect. 2.1 and the training of the denoising network in Sect. 2.2. We introduce two different sampling schemes in Sect. 2.3 and different parameterizations of the NN output in Sect. 2.4.





## 2.1 Gaussian diffusion process

The diffusion process is defined in terms of intermediate latent (noised) states $\boldsymbol{z}_\tau$ with $\tau \in [0, T]$ as discrete pseudo-time steps.
As these latent states are noised state samples, they still lay in state space, and we define $\boldsymbol{z}_0 = \boldsymbol{x}$.

The diffusion process progressively adds small Gaussian noise, $\boldsymbol{\epsilon}_\tau \sim \mathcal{N}(0, \sigma_\tau^2 \mathbf{I})$, to the states, where $\sigma_\tau$ describes the amplitude of the added noise at pseudo-time $\tau$. Since the noise accumulates, the variance of the states would increase with pseudo-time. Instead, we use here a *variance-preserving* formulation, where the signal is progressively replaced by noise. The signal magnitude is decreased in pseudo-time $1 \geq \alpha_{\tau-1} > \alpha_\tau \geq 0$ with $\sigma_\tau = \sqrt{1 - \alpha_\tau}$, such that the variance remains the same for all pseudo-time steps, if the state samples are normalized. The function that defines the signal magnitude as function of the pseudo-time step is called noise scheduler. To simplify the derivation in the following, we assume a given noise scheduling and show the definitions of the two used noise schedulers in Appendix C and refer to Sect. 3.2 of Nichol and Dhariwal (2021) for a more detailed discussion.

The transition of the latent state $\boldsymbol{z}_\tau$ from pseudo-time $\tau - 1$ to pseudo-time $\tau$ with $\tau \geq \tau - 1$ is then given as

$$q(\boldsymbol{z}_\tau \mid \boldsymbol{z}_{\tau-1}) = \mathcal{N}\left(\sqrt{\alpha'_\tau} \boldsymbol{z}_{\tau-1}, (1 - \alpha'_\tau)\mathbf{I}\right), \tag{3}$$

with the relative signal magnitude $\alpha'_\tau = \frac{\alpha_\tau}{\alpha_{\tau-1}}$. Using the additive property of Gaussian distributions, the distribution of the latent state $\boldsymbol{z}_\tau$ at step $\tau$ can be directly defined given a state sample $\boldsymbol{x}$ and a signal magnitude $\alpha_\tau$,

$$q(\boldsymbol{z}_\tau \mid \boldsymbol{x}) = \mathcal{N}\left(\sqrt{\alpha_\tau} \boldsymbol{x}, (1 - \alpha_\tau)\mathbf{I}\right). \tag{4}$$

Setting the signal magnitude at the last step near zero, $\alpha_T \approx 0$, the signal vanishes and the latent states converge towards a normal Gaussian distribution, which then also defines our prior distribution,

$$q(\boldsymbol{z}_T \mid \boldsymbol{x}) = p(\boldsymbol{z}_T) = \mathcal{N}(\mathbf{0}, \mathbf{I}). \tag{5}$$

Given the transition distribution from Eq. (3), the joint distribution of the trajectory for the diffusion process forward in pseudo-time reads

$$q(\boldsymbol{z}_{1:T} \mid \boldsymbol{x}) = \prod_{t=0}^{T} q(\boldsymbol{z}_\tau \mid \boldsymbol{z}_{\tau-1}). \tag{6}$$

The NN is then trained to reverse a single step of this trajectory, such that we can start from the prior distribution, Eq. (5), and generate the trajectory without needing access to the state sample $\boldsymbol{x}$, as we will see in the following section.

## 2.2 Training procedure

During training, we sample a latent state from the trajectory by drawing a state $\boldsymbol{x} \sim p_{\mathsf{data}}(\boldsymbol{x})$, noise $\boldsymbol{\epsilon} \sim \mathcal{N}(0, \mathbf{I})$, and a pseudo-time $\tau$ which specifies the signal magnitude $\alpha_t$. Making use of the reparameterization property of Gaussian distributions, we can write the latent state drawn from its distribution $q(\boldsymbol{z}_\tau \mid \boldsymbol{x})$ as

$$\boldsymbol{z}_\tau = \sqrt{\alpha_\tau} \boldsymbol{x} + \sqrt{(1 - \alpha_\tau)} \boldsymbol{\epsilon}. \tag{7}$$



To describe the analytical denoising step from $\tau$ to $\tau-1$, we use Bayes' theorem given the definition of the diffusion process, as defined in Eq. (3) and Eq. (4), and a known state sample $\boldsymbol{x}$,

$$q(\boldsymbol{z}_{\tau-1} \mid \boldsymbol{z}_\tau, \boldsymbol{x}) = \mathcal{N}\big(\mu(\boldsymbol{z}_\tau, \boldsymbol{x}), \Sigma_\tau\big) \tag{8a}$$

with  $\mu(\boldsymbol{z}_\tau, \boldsymbol{x}) = \dfrac{\sqrt{\alpha'_\tau}(1-\alpha_{\tau-1})}{1-\alpha_\tau}\boldsymbol{z}_\tau + \dfrac{\sqrt{\alpha_{\tau-1}}(1-\alpha'_\tau)}{1-\alpha_\tau}\boldsymbol{x}$ (8b)

and  $\Sigma_\tau = \dfrac{(1-\alpha'_\tau)(1-\alpha_{\tau-1})}{1-\alpha_\tau}\mathbf{I}$ (8c)

The diffusion and the denoising process are defined over several signal magnitudes. We train one NN for all signal magnitudes and use the pseudo-time as additional input into the NN. Here, we parameterize the NN to predict the drawn noise, $\widehat{\boldsymbol{\epsilon}}_{\boldsymbol{\theta}}(\boldsymbol{z}_\tau, \tau) = f_{\boldsymbol{\theta}}(\boldsymbol{z}_\tau, \tau)$ based on the current latent state and pseudo-time. We introduce other parameterizations in Sect. 2.4.

To approximate the analytical denoising step from Eq. (8a) with the NN, we have to specify the unknown state $\boldsymbol{x}$ by the NN output. Using the predicted noise, the latent state can be directly projected to the state by Tweedie's formula (Efron, 2011),

$$\widehat{\boldsymbol{x}}_{\boldsymbol{\theta}}(\boldsymbol{z}_\tau, \tau) = \mathbb{E}\Big[p(\boldsymbol{x} \mid \boldsymbol{z}_\tau)\Big] = \frac{1}{\sqrt{\alpha_\tau}}\Big(\boldsymbol{z}_\tau - \sqrt{(1-\alpha_\tau)}\,\widehat{\boldsymbol{\epsilon}}_{\boldsymbol{\theta}}(\boldsymbol{z}_\tau, \tau)\Big). \tag{9}$$

Replacing the state by this prediction in the mean function, Eq. (8b), we have completely specified the denoising distribution with predicted quantities,

$p_{\boldsymbol{\theta}}(\boldsymbol{z}_{\tau-1} \mid \boldsymbol{z}_\tau) = q\big(\boldsymbol{z}_{\tau-1} \mid \boldsymbol{z}_\tau, \widehat{\boldsymbol{x}}_{\boldsymbol{\theta}}(\boldsymbol{z}_\tau, \tau)\big).$ (10)

The goal is to make the approximation, Eq. (10), as close as possible to the analytical denoising step from Eq. (8a) using the Kullback-Leibler divergence between the approximation and the analytical step,

$$D_{KL}\big(q(\boldsymbol{z}_{\tau-1} \mid \boldsymbol{z}_\tau, \boldsymbol{x})\|p_{\boldsymbol{\theta}}(\boldsymbol{z}_{\tau-1} \mid \boldsymbol{z}_\tau)\big)$$
$$= \mathbb{E}_{\boldsymbol{z}_{\tau-1}\sim q(\boldsymbol{z}_{\tau-1}|\boldsymbol{z}_\tau,\boldsymbol{x})}[\log q(\boldsymbol{z}_{\tau-1} \mid \boldsymbol{z}_\tau, \boldsymbol{x}) - \log p_{\boldsymbol{\theta}}(\boldsymbol{z}_{\tau-1} \mid \boldsymbol{z}_\tau)]. \tag{11}$$

By definition, the covariance of the approximated and analytical denoising step match, and the Kullback-Leibler divergence, Eq. (11), reduces to a mean-squared error loss between the predicted noise and the used randomly drawn noise,

$$D_{KL}\big(q(\boldsymbol{z}_{\tau-1} \mid \boldsymbol{z}_\tau, \boldsymbol{x})\|p_{\boldsymbol{\theta}}(\boldsymbol{z}_{\tau-1} \mid \boldsymbol{z}_\tau)\big) \propto \frac{1}{w(\tau)}\|\widehat{\boldsymbol{\epsilon}}(\boldsymbol{z}_\tau, \tau) - \boldsymbol{\epsilon}\|_2^2, \tag{12}$$

with weighting factor $w(\tau) = \frac{\alpha_\tau}{1-\alpha_\tau}$, the signal-to-noise ratio. In practise, this weighting factor is neglected (Ho et al., 2020), which leads to a simplified loss function.

The NN is trained for all pseudo-time steps to achieve its optimal parameters $\boldsymbol{\theta}^\star$. For a single training step, we minimize the expectation of the simplified loss function of Eq. (12),

$$\boldsymbol{\theta}^\star = \arg\min_{\boldsymbol{\theta}} \mathbb{E}_{\boldsymbol{x}\sim p_{\text{data}}(\boldsymbol{x}), \boldsymbol{\epsilon}\sim\mathcal{N}(\mathbf{0},\mathbf{I}), \tau\sim\mathcal{U}(1,T)}\Big[\|\widehat{\boldsymbol{\epsilon}}(\sqrt{\alpha_\tau}\boldsymbol{x} + \sqrt{(1-\alpha_\tau)}\boldsymbol{\epsilon}, \tau) - \boldsymbol{\epsilon}\|_2^2\Big], \tag{13}$$

where $\mathcal{U}(1,T)$ is a uniform distribution with 1 and $T$ as bounds. Eq. (12) can be derived from the so-called evidence lower bound (Kingma et al., 2021; Luo et al., 2023), and we optimize a weighted lower bound to the unknown distribution $p_{\text{data}}(\boldsymbol{x})$

of the states on the attractor with Eq. (13). Consequently, we can expect that the better the prediction of the NN, the nearer the generated state samples to the attractor of the dynamical system are.





## 2.3 Sampling from the denoising process

After training the NN, we can use it to sample from the denoised state trajectory distribution $p_{\boldsymbol{\theta}}(\boldsymbol{z}_{0:T})$, defined in Eq. (2). To sample from the distribution, we can start sampling from the prior distribution $\boldsymbol{\epsilon}_T \sim \mathcal{N}(\mathbf{0},\mathbf{I})$, and then sample for each subsequent denoising step from Eq. (10),

$$\boldsymbol{z}_{\tau-1}(\boldsymbol{z}_\tau,\tau) = \frac{\sqrt{\alpha'_\tau}(1-\alpha_{\tau-1})}{1-\alpha_\tau}\boldsymbol{z}_\tau + \frac{\sqrt{\alpha_{\tau-1}}(1-\alpha'_\tau)}{1-\alpha_\tau}\widehat{\boldsymbol{x}}_{\boldsymbol{\theta}}(\boldsymbol{z}_\tau,\tau) + \sqrt{\frac{(1-\alpha'_\tau)(1-\alpha_{\tau-1})}{1-\alpha_\tau}}\boldsymbol{\epsilon}, \qquad \boldsymbol{\epsilon}\sim\mathcal{N}(\mathbf{0},\mathbf{I}). \qquad (14)$$

This sampling process is inherently stochastic and in the following called *denoising diffusion probabilistic model* (DDPM, Ho et al., 2020).

To reduce the magnitude of randomness during training, the sampling process can be made deterministic (Song et al., 2020a, 2021). This deterministic sampling scheme is called *denoising diffusion implicit models* (DDIM, Song et al., 2020a), and its only source of randomness is in the sampling from the prior distribution. The marginal distribution of the generated state samples remains the same, and the model can be still trained by the same loss function as defined in Eq. (13).

In DDIMs, the noise $\boldsymbol{\epsilon}_\tau$ drawn from a Gaussian distribution is replaced by the predicted noise from the NN $\widehat{\boldsymbol{\epsilon}}(\boldsymbol{z}_\tau,\tau)$, also used to predict the state $\widehat{\boldsymbol{x}}_{\boldsymbol{\theta}}(\boldsymbol{z}_\tau,\tau)$. We can introduce an additional factor $\eta\in[0,1]$, which determines the randomness in the sampling process. Given this factor, we can sample from the denoising steps as follows,

$$\boldsymbol{z}_{\tau-1} = \sqrt{\alpha_{\tau-1}}\widehat{\boldsymbol{x}}_{\boldsymbol{\theta}}(\boldsymbol{z}_\tau,\tau) + \sqrt{(1-\alpha_{\tau-1}-\sigma_\tau^2)}\widehat{\boldsymbol{\epsilon}}(\boldsymbol{z}_\tau,\tau) + \sigma_\tau\boldsymbol{\epsilon}, \qquad \boldsymbol{\epsilon}\sim\mathcal{N}(\mathbf{0},\mathbf{I}) \qquad (15a)$$

$$\sigma_\tau = \eta\sqrt{\frac{1-\alpha_{\tau-1}}{1-\alpha_\tau}}\sqrt{1-\frac{\alpha_\tau}{\alpha_{\tau-1}}}. \qquad (15b)$$

The factor interpolates between purely deterministic sampling with DDIMs, $\eta=0$, and fully stochastic samples, $\eta=1$. Sampling with Eq. (14) has an even larger randomness than $\eta=1$. Note, sampling with $\eta=1$ and the earlier introduced sampling with Eq. (14) are both DDPMs. For simplicity, we refer to Eq. (14) as DDPM, whereas we call sampling with $\eta=1$ a stochastic DDIM scheme. Throughout the manuscript, we normally sample with Eq. (14), whereas we also perform experiments with DDIM schemes.

## 2.4 Output parameterizations

Usually, the output of the NN is parameterized as prediction $\widehat{\boldsymbol{\epsilon}}_{\boldsymbol{\theta}}(\boldsymbol{z}_\tau,\tau) = f_{\boldsymbol{\theta}}(\boldsymbol{z}_\tau,\tau)$ of the noise (Ho et al., 2020). Here, we introduce two additional parameterizations and discuss their advantages and disadvantages. In our implementation, a different parameterization also changes the loss function for the NN. The change in the loss function modifies then the implied weighting of the Kullback-Leibler divergence in Eq. (12), as shown in Fig. 2a.

In Eq. (9), we have defined the predicted state as a function of the predicted noise. Instead, we can also directly predict the state $\widehat{\boldsymbol{x}}_{\boldsymbol{\theta}}(\boldsymbol{z}_\tau,\tau) = f_{\boldsymbol{\theta}}(\boldsymbol{z}_\tau,\tau)$. With this parameterization, we minimize the mean-squared error of the predicted state to the true state during training, which gives a constant weighting of the Kullback-Leibler divergence, shown as red curve in Fig. 2a.





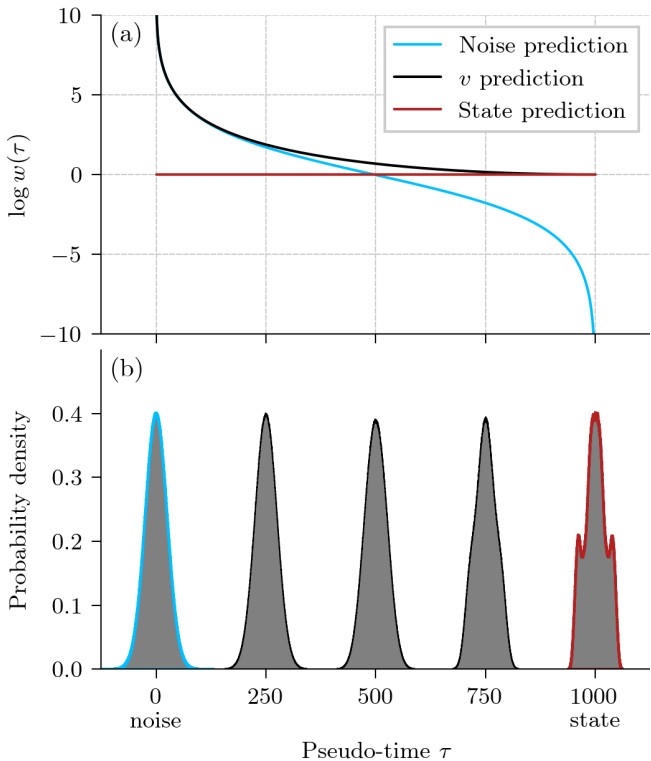

**Figure 2.** Comparison between different output parameterizations for denoising diffusion models. (a) The logarithm of the weighting $\log w(\tau)$ in the Kullback-Leibler divergence in Eq. (12) as a function of pseudo-time step $\tau$, with a cosine noise scheduling (see also Appendix C). The weighting of the noise prediction corresponds to the signal-to-noise ratio (SNR), the weighting of the $v$ prediction to SNR+1, and the weighting for the state prediction is constant. (b) Empirical probability density functions (PDF) that are targetted during a $v$ prediction for the $x$-component of the Lorenz 1963 model and several pseudo-time steps. For a pseudo-time step of 0, the PDF corresponds to the prior, and for a pseudo-time step of $T = 1000$, to the state.

The NN is trained to split the signal and noise from a given latent state. There, we could either directly predict the state (signal) or the noise. Nonetheless, we can alternatively define as target a combination of both (Salimans and Ho, 2022),

$$\boldsymbol{v}(\boldsymbol{x}, \boldsymbol{\epsilon}, \tau) = \sqrt{\alpha_\tau}\boldsymbol{\epsilon} - \sqrt{(1 - \alpha_\tau)}\boldsymbol{x}. \tag{16}$$

Predicting $\widehat{\boldsymbol{v}}(\boldsymbol{x}, \boldsymbol{\epsilon}, \tau) = f_{\boldsymbol{\theta}}(\boldsymbol{z}_\tau, \tau)$ and minimizing the mean-squared error between prediction and true $\boldsymbol{v}$ interpolates the weighting between noise ($\tau = 0$) and state ($\tau = 1000$) prediction, shown as black curve in Fig. 2a.

Since the state is needed in the denoising step, Eq. (8a), predicting the state is a straightforward parameterization for training and applying the NN. However, the distribution of the state might be non-Gaussian and multimodal, as shown in Fig. 2b, such that the Gaussian approximation for the loss function could be inadequate. By contrast, the noise is drawn from a Gaussian distribution and the mean-squared error is statistically speaking the correct loss function. Additionally, Ho et al. (2020) have shown that predicting the noise leads to better results than directly predicting the state.



However, predicting the noise can be highly unstable for low signal-to-noise ratios (Salimans and Ho, 2022), the $\boldsymbol{v}$-prediction weights the loss function differently and circumvents these instabilities. As the target is a combination between state and noise, the target distribution shifts from non-Gaussian state prediction for small signal magnitudes to Gaussian noise prediction for large signal magnitudes, Fig. 2b. Consequently, the use of the $\boldsymbol{v}$-prediction can be advantageous for the training stability and

200 the loss function, which may improve the performance of the DDMs.

## 3 Downstream tasks for unconditional denoising diffusion models

In this manuscript, we train DDMs to generate states that should lay on the attractor of the dynamical system. As training data, we integrate the equations that define the dynamical system to produce a long state trajectory. Using each state of the trajectory at discrete time as training sample, the DDM is trained to unconditionally generate state samples. We reason that the DDM

must have learned an internal representation of the attractor. In Appendix F, we analyze the extracted features of the denoising NN and show that this representation is entangled; we need all extracted features to extract information about the attractor. In the following, we explain two different approaches on how the unconditional DDM can be used for downstream tasks, other than pure state generation.

First, in Sect. 3.1, we demonstrate the use of the denoising NN for transfer learning; we fine-tune it for surrogate modelling.

Secondly, in Sect. 3.2, we generate with the DDM a state ensemble from a deterministic forecast run.

### 3.1 Transfer learning from the denoising neural network

As schematically shown in Fig. 3, our general idea of transfer learning the NN is to remove its last layer. This last layer combines the extracted features $\phi(\boldsymbol{z}_\tau, \tau)$ at a specific pseudo-time step $\tau$ by the weights $\mathbf{W}$ and the bias $\boldsymbol{\beta}$ to the NN output $f_{\boldsymbol{\theta}}(\boldsymbol{z}_\tau, \tau) = \mathbf{W}^\top \phi(\boldsymbol{x}_\tau, \tau) + \boldsymbol{\beta}$.

Since the noised states of the DDM remain in state space, the network can be easily applied to cleaned states, instead of working with noised states. Keeping the pseudo-time step fixed, we can extract features $\phi(\boldsymbol{x}_t)$ from a given state $\boldsymbol{x}_t$ at a physical time step $t$ with the NN by removing its last layer. For the task of surrogate modelling, we regress these extracted features to the next state $\boldsymbol{x}_{t+1}$, one time step later. As tuning parameter for the feature extractor, we can select the pseudo-time step and concatenate features at multiple pseudo-time steps.

The increasing noise magnitude with pseudo-time forces the network to extract fine features for small pseudo-time steps and coarse features for large pseudo-time steps. We visualize this in Fig. 4a, where we project the activation of an arbitrary neuron onto the $x$-$z$ plane of the Lorenz 1963 model. The model extracts different features at different pseudo-time steps, even for smaller pseudo-time steps like $\tau = 0$ and $\tau = 200$. Nevertheless, the NN extracts more complex feature for such smaller pseudo-time steps, whereas the extracted features are more linearly separated for larger pseudo-time steps.

To test the linearity of the extracted features, we fit a linear regression from state space $\boldsymbol{x}_t$ to extracted feature space $\phi(\boldsymbol{x}_t, \tau)$, such that the following relation should approximatively hold,

$$\phi(\boldsymbol{x}_t, \tau) \approx \mathbf{W}^\top \boldsymbol{x}_t + \boldsymbol{\beta}. \tag{17}$$





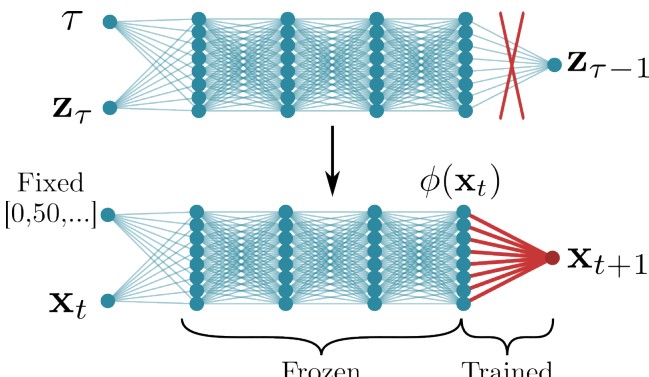

**Figure 3.** Schema for fine-tuning a denoising diffusion model for surrogate modelling. In the denoising network, the last layer is removed, and the remaining network is used as feature extractor, extracting features at different pseudo-time steps. These features $\phi(\boldsymbol{x}_t)$ are then used as input for a linear regression or a small NN to predict the next state $\boldsymbol{x}_{t+1}$ based on the current state. The biggest part of the NN is frozen and remains untouched during this transfer learning procedure.

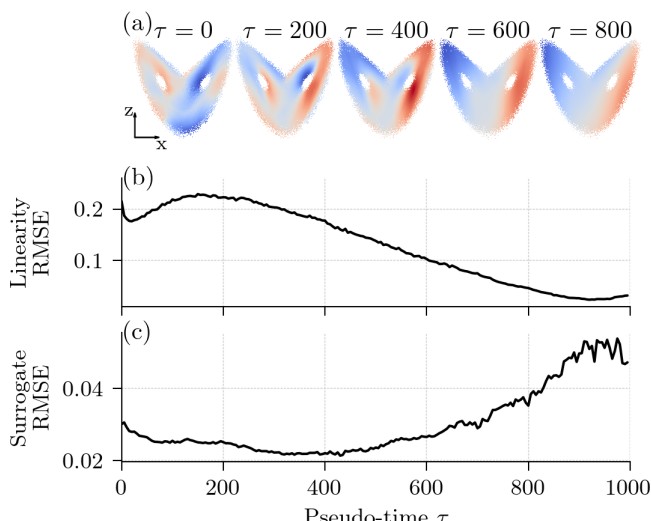

**Figure 4.** The learned representation of a pre-trained diffusion model with a $v$ parameterization and a cosine noise scheduling, depending on the pseudo-time step. (a) Extracted features projected onto the $x$-$z$ plane for an arbitrary neuron and five different pseudo-time steps. The colormap is proportional to the anomaly of the neuron's excitation. The anomaly is positive for regions in red and negative for regions in blue. (b) The RMSE of a linear regression from the three-dimensional state space to the extracted 256-dimensional feature space, see also Eq. (17). (c) The nRMSE of a linear regression from the extracted features to the state dynamics after 10 integration steps (0.1 MTU), see also Eq. (18).

The larger the pseudo-time step, the smaller the error of the linear regression (Fig. 4b), and the better the features can be linearly predicted from the state space. The network extracts the most non-linear features for an intermediate pseudo-time step around $\tau = 200$. These results confirm the visual results in Fig. 4a.



The ordinary differential equations of the Lorenz 1963 system include second-order polynomial terms, and, integrated in time, the influence of the non-linearities in the system increases with lead time. To learn a surrogate model for multiple integration time steps, we need to extract non-linear features. Consequently, we can expect that if the features are more non-linear, they can be better used for surrogate modelling.

To test this hypothesizes, we fit a linear regression from feature space to the dynamics of the model after 10 integration time steps, $\Delta t = 0.1\,\mathrm{MTU}$ (model time units), such that the following relation should approximatively hold,

$$\boldsymbol{x}_{t+\Delta t} - \boldsymbol{x}_t \approx \mathbf{W}^\top \phi(\boldsymbol{x}_t, \tau) + \boldsymbol{\beta}, \tag{18}$$

The smaller its error, the more useful are the features for surrogate modelling.

As the dynamics are non-linear, we also need non-linear features for surrogate modelling. Consequently, the larger the pseudo-time step, the less the predictions can explain the dynamics of the system, and the larger the regression error, as can be seen in Fig. 4c. The features most linearly linked to the dynamics are around an intermediate pseudo-time step $\tau = 400$. As different features at different pseudo-time steps are extracted, we propose to extract features at multiple pseudo-time steps between $\tau = 0$ and $\tau = 600$. These features are concatenated as predictors in a linear regression or small NN for surrogate modelling.

## 3.2 Ensemble generation by diffusing and denoising a deterministic forecast run

Beside the feature space of the denoising NN, the latent space also encodes useful information for other tasks than the network was trained on (e.g. Song et al., 2020a). In a second approach, we use the latent space to generate a state ensemble from a deterministic forecast run. This approach resembles the approach of SDEdit (Meng et al., 2022) to guide the editing of images with DDMs.

Our idea is to partially diffuse a deterministic forecast until a given signal magnitude $\alpha_\tau$ is reached and to reconstruct an ensemble out of the latent space. The diffusion process from state to latent state is intrinsically stochastic and, thus, a one-to-many mapping. Taking samples in the latent space, we reconstruct an ensemble by iteratively applying the denoising network for the same number of pseudo-time steps as used to diffuse the deterministic forecast, as schematically shown in Fig. 5a.

The denoising network is state-dependent, which makes also the DDM for ensemble generation state-dependent, as shown in Fig. 5b. Trained for state generation only, the DDM has never seen any time-dependent relationships between samples. Consequently, the relationship between samples is purely induced by the climatology, and the state-dependency hardly translates into a flow-dependency.

The denoising process is trained to generate states on the attractor of the dynamical system. The chosen pseudo-time step consequently controls the sampled portion of the attractor. As we will see later, the bigger the Because of the state-dependency, the resulting distribution is implicitly represented by the ensemble and could extend beyond a Gaussian assumption. We formalize the ensemble generation and the implicit distribution representation in Appendix B, showing its connection to a Bayesian framework.





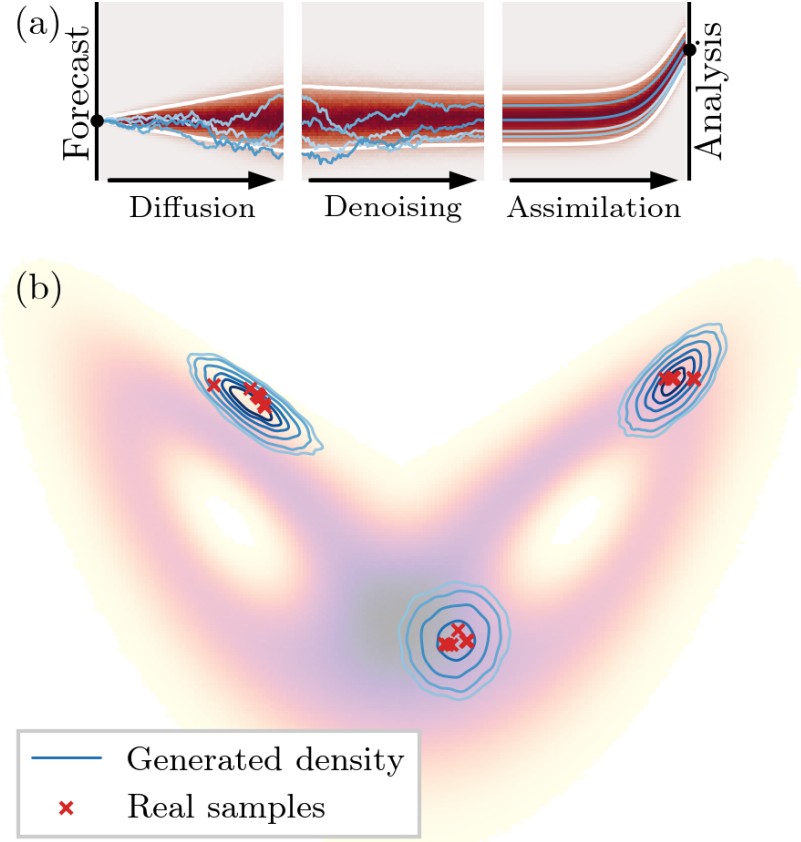

**Figure 5.** Schematic visualization of the ensemble generation with denoising diffusion models for data assimilation. (a) The deterministic run is diffused for a given number of pseudo-time steps towards the prior distribution $\mathcal{N}(\mathbf{0}, \mathbf{I})$ and then reconstructed with the pre-trained NN. As the diffusion process is stochastic, we can generate several latent states from which we can reconstruct an ensemble. This ensemble is used in an ensemble optimal interpolation to correct the deterministic run towards given observations. The black dot in the beginning and at the end correspond to the forecasted and corrected deterministic run, the blue lines depict four single ensemble members, the white lines to the 5th and 95th percentiles, and the red background shows the density. For graphical purpose, the assimilation step has a decreasing observation error to its final value to depict a flow from prior to posterior distribution. (b) Three examples of generated ensembles on the attractor of the Lorenz 1963 system, projected onto the $x$-$z$ plane. The generated densities are estimated based on ensembles generated with a denoising diffusion model. The red cross correspond to ensemble members from a tuned ensemble Kalman filter system for the same time as the generated densities. The shading in the background is the density function of the full Lorenz 1963 system.

The bigger the pseudo-time step, the smaller the signal magnitude, and the more diffused is the deterministic run, which controls the degree of uncertainty in the ensemble. For a very small pseudo-time step with a signal magnitude near one, $\alpha_\tau \approx 1$, almost no noise would be added, and we would end up with a very small ensemble spread. For a large pseudo-time step with a signal magnitude near zero, $\alpha_\tau \approx 0$, almost all data would be replaced by noise in the latent state; the generated



ensemble would correspond to a climatological ensemble. In general, the choice of the pseudo-time step is similar to the covariance inflation factor in an ensemble data assimilation system.

We test this ensemble generation approach in data assimilation with an ensemble Kalman filter. In fact, this methodology is an ensemble optimal interpolation approach (EnOI, Evensen, 2003; Oke et al., 2002). Instead of specifying an explicit covariance or providing states drawn from a climatology, the samples generated with the DDM implicitly represent the prior distribution for the data assimilation.

## 4   Experiments

We showcase the potential of DDMs for representation learning in geoscientific systems with three different type of experiments. In the state generation experiments (Sect. 4.1), we establish the methodology of DDMs. We test different settings for the denoising network and compare these results to the best practices in computer vision for image generation. Afterwards, two downstream applications are built around the best-performing denoising network. In the transfer learning experiments (Sect. 4.2), we use the pre-trained denoising network as feature extractor for surrogate modelling of the Lorenz 1963 system, see also Sect. 3.1. In the ensemble generation experiment (Sect. 4.3), the DDM is combined with an ensemble optimal interpolation to assimilate observations into a deterministic forecast. Using these data assimilation experiments, we can assess how well the DDM can generate an ensemble out of deterministic forecasts, see also Sect. 3.2.

We perform all experiments with the Lorenz 1963 model (Lorenz, 1963). Its dynamical system has three variables $x$, $y$, $z$ and is defined by the following set of ordinary differential equations, where we use the standard parameters,

$$\frac{dx}{dt} = \sigma(y - x), \qquad\qquad \sigma = 10 \qquad\qquad (19\text{a})$$

$$\frac{dy}{dt} = x(\rho - z) - y, \qquad\qquad \rho = 28 \qquad\qquad (19\text{b})$$

$$\frac{dz}{dt} = xy - \beta z, \qquad\qquad \beta = \frac{8}{3}. \qquad\qquad (19\text{c})$$

The chosen parameters induce a chaotic behaviour with an error doubling time of $0.78\,\text{MTU}$ (model time units). We integrate the dynamical system with a fourth-order Runge-Kutta integrator and an integration time step of $0.01\,\text{MTU}$.

We base our experiments on an ensemble of 33 trajectories (16 for training, 1 for validation, and 16 for testing), initialized with random states, sampled from $\mathcal{N}(\mathbf{0}, (0.001)^2\mathbf{I})$. The first $1 \times 10^5$ integration steps are omitted as spin-up time. We integrate the system with additional $1 \times 10^6$ steps to generate the states needed for training, validation, and testing. This way we generate $1.6 \times 10^7$ samples for training, $1 \times 10^6$ for validation, and $1.6 \times 10^7$ for testing. This large number of samples allows us to test settings without being constrained by data. Before training, the data is normalized by the mean and standard deviation estimated based on the training dataset. The code is developed in Python (Van Rossum, 1995), using PyTorch (Paszke et al., 2019), and PyTorch lightning (Falcon et al., 2020), and is publicly available under: https://github.com/cerea-daml/ddm-attractor.





## 4.1 State generation

As denoising network, we use a ResNet-like architecture (He et al., 2015) with fully-connected layers, for more information see Appendix D1. To condition the network on the pseudo-time step, we encode the discrete pseudo-time ($\tau \in [0, 1000]$) by a sinusoidal encoding (Vaswani et al., 2017). The encoded pseudo-time modulates via a linear function to the scale and shifting
parameters of the layer normalizations in the residual layers. In total, the denoising network has $1.2 \times 10^6$ parameters, a very large number of parameters for the Lorenz 1963 system. However, we are in a training data regime with a very large number of samples, rolling out the state generation experiments without worrying about underfitting of the network.

The networks are trained with the Adam (Kingma and Ba, 2017, $\gamma = 3 \times 10^{-4}$) optimizer for $100$ epochs with a batch size of $16384$. To reduce the amount of randomness in the results, each experiment is performed ten times with different seeds,
which randomize the initial weights for the neural network, the order of the samples within one epoch, and the noise added to the samples during training.

If not differently specified, we sample $1 \times 10^6$ states with the DDPM scheme, defined in Eq. 14, for $T = 1000$ pseudo-time steps, as the networks are trained for. These generated states are compared to the testing samples using five different metrics, for exact definitions see also Appendix E.
We compare how near the generated states are to the attractor by using the Hellinger distance $H$ between the generated state distribution and the testing state distribution (Arnold et al., 2013; Gagne II et al., 2020); to estimate the distance, we discretize the state into cubes (Scher and Messori, 2019). The Hellinger distance is bounded $0 \le H \le 1$ with $H = 0$, if the distributions perfectly correspond to each other, and $H = 1$ if there is no overlap.

To measure the perceptual quality of the generated states, we adapt the Fréchet inception distance to our Lorenz 1963
settings, in the following called Fréchet surrogate distance (FSD). Replacing the inception network, we estimate the Fréchet distance in feature space spanned by a dense neural network with two hidden layers, trained for surrogate modelling. The smaller the distance, the better match the statistics of the generated state distribution to the testing state distribution in feature space. Heusel et al. (2017) have shown that the Fréchet inception distance is consistent with human judgement on disturbed image data.
We additionally compute the squared distance of the nearest neighbor between generated states and the testing samples, either as expectation over the generated states $\overline{d}_{\text{gen}}$, or as expectation over the testing states $\overline{d}_{\text{test}}$. To evaluate rare events, we use a peak-over-threshold metric (POT). We use the first and 99th percentile from the testing dataset such that $2\,\%$ of the generated samples should lay in average below and above the lower and upper threshold, respectively.

### 4.1.1 Results

In our simplified formulation, changing the parameterization of the neural network output changes the loss function and the weighting of the Kullback-Leibler divergence, as explained in Sect. 2.4. The weighting is additionally influenced by the chosen noise scheduling, here either a linear scheduler or a cosine scheduler (Nichol and Dhariwal, 2021), both defined in Appendix C. In Tab. 1, we compare the output parameterizations and noise scheduler in terms of the resulting generative quality.



**Table 1.** Performance of samples generated with different output parametrizations and noise schedulers compared to the test dataset. All samples are generated for 1000 pseudo-time steps and a DDPM scheme. The first two entries are values for the training and validation data. The scores are averaged across ten neural networks trained with different random seeds and multiplied by the number in brackets. Downward arrows show the lower the score, the better. Bold values indicate the best performing method for a given column, and the velocity parameterization with a cosine noise scheduler is used in subsequent experiments.

| Parametrization | Scheduler | $H(\times 10^2) \downarrow$ | $FSD(\times 10^3) \downarrow$ | $\overline{d}_{gen}(\times 10^6) \downarrow$ | $\overline{d}_{test}(\times 10^5) \downarrow$ | POT |
|---|---|---|---|---|---|---|
| Training | - | 1.28 | 0.23 | 0.87 | 0.09 | 2.00 |
| Validation | - | 3.11 | 0.26 | 0.79 | 1.40 | 2.01 |
| State $x$ | Linear | 17.77 | 159.08 | 125.62 | 13.29 | 1.73 |
| State $x$ | Cosine | 14.76 | 170.56 | 66.94 | 8.88 | 1.90 |
| Noise $\epsilon$ | Linear | 4.64 | 7.27 | 8.27 | 1.69 | **1.92** |
| Noise $\epsilon$ | Cosine | **4.03** | 5.75 | 4.58 | **1.53** | 1.89 |
| Velocity $v$ | Linear | 4.54 | 7.66 | 8.09 | 1.69 | 1.88 |
| **Velocity $v$** | **Cosine** | **4.00** | **5.17** | **4.50** | **1.52** | 1.91 |

The noise $\epsilon$ and velocity $v$ parameterizations results into the best scores. Additionally, a cosine noise scheduler improves almost all scores compared to a linear scheduler. During training (not shown), we have experienced that the velocity $v$ parameterization is more stable and converges faster than the noise $\epsilon$ parameterization. These results confirm results from image generation, where a velocity $v$ parameterization has been introduced to stabilize the training of the neural network (Salimans and Ho, 2022). Hence, we recommend a velocity $v$ parameterization and a cosine noise scheduler as default combination for training DDMs. All following results are consequently derived based on this configuration (also marked as bold in Table 1).

Analyzing the scores for the peak over threshold metric (POT), the DDMs are slightly underdispersive. However, the better the model, the better the coverage. Furthermore, the generated states visually cover the testing samples, in terms of two-dimensional projections as shown in Fig. 6a–f. Comparing the one-dimensional marginal empirical probability density functions in Fig. 6g–i, the generated samples are almost indistinguishable from the true samples, even in their extreme values. Taking Table 1 and Fig. 6 into account, DDMs can generate state samples very similar to those drawn from the attractor of the dynamical system.

Since the denoising neural network must be iteratively applied, generating samples with DDMs can be slow, especially for high-dimensional states. Trained with 1000 pseudo-time steps, DDMs can generate samples by skipping steps to speed up the generation process. We evaluate the effect of fewer generation steps in Tab. 2, where the generation quality is measured in terms of Hellinger distance. As the DDIM sampling scheme have been introduced for data generation with fewer steps (Song et al., 2020a), we additionally evaluate the impact of the additional noise during sampling.

For all sampling schemes, the quality of the generated samples improves with the number of pseudo-time steps. However, the improvements between 100 pseudo-time steps and 1000 pseudo-time steps are small compared to differences from different





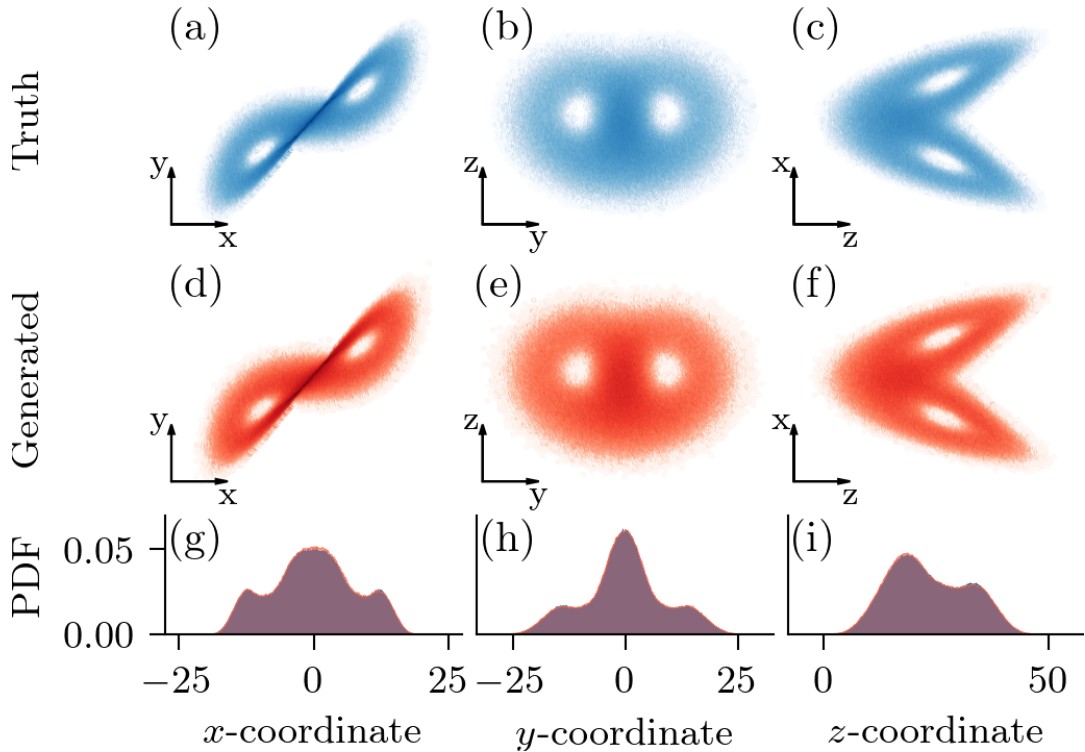

**Figure 6.** (a)–(c), true samples from the testing dataset, projected into the depicted two dimensions. (d)–(f), generated samples for a denoising diffusion model with $v$ parameterization, cosine scheduler, and 1000 pseudo-time steps with a DDPM scheme, projected into the depicted two dimensions. (g)–(i), marginal one-dimensional empirical probability density functions (PDF) for the samples from the testing dataset (blue) and the generated samples (red). The overlap in the marginal distributions results into the magenta-like colour.

**Table 2.** Hellinger distance between the generated state distribution and the test state distribution for different sampling schemes and generation steps; the lower the distance, the better. For visual convenience, the Hellinger distances are multiplied by $1 \times 10^2$. $\eta$ corresponds to the noise magnitude in the sampling scheme, as defined in Eq. 15b. The distances are averaged across ten different random seeds. Bold numbers indicate the best distance for a given number of time steps, and the DDPM sampling is used in all other experiments.

| Scheme | 10 | 20 | 50 | 100 | 1000 |
|---|---|---|---|---|---|
| DDIM ($\eta = 0.0$) | **24.09** | **14.41** | **7.18** | **5.01** | 3.94 |
| DDIM ($\eta = 0.2$) | 24.16 | 14.45 | **7.18** | **5.00** | 3.90 |
| DDIM ($\eta = 0.5$) | 24.54 | 14.75 | 7.28 | **5.01** | **3.86** |
| DDIM ($\eta = 1.0$) | 27.55 | 16.68 | 8.19 | 5.43 | 3.90 |
| **DDPM** | 27.50 | 16.60 | 8.15 | 5.40 | 3.97 |

output parameterizations and the noise schedulers. With a smaller computational budget to generate data, a DDM can generate data with a still an acceptable quality in fewer pseudo-time steps.



Similar to the results found for image generation (Song et al., 2020a), reducing the added noise during generation of the state samples can improve the quality of the generated samples for a smaller number of time steps than trained for. The deterministic DDIM with $\eta = 0.0$ is the best performing sampling scheme for a smaller number of pseudo-time steps than 50. However, when the full 1000 pseudo-time steps are used for data generation, almost no differences are left between the different sampling schemes.

**4.2  Surrogate modelling**

In this next step, we fine-tune a DDM as feature extractor for surrogate modelling. As a reminder, the model is pre-trained with the velocity $\boldsymbol{v}$ parameterization and a cosine noise scheduling. Based on the initial state $\boldsymbol{x}_t$ at time $t$, we want to predict the state $\boldsymbol{x}_{t+\Delta t}$ for a lead time $\Delta t = 0.1\,\mathrm{MTU}$, a mildly non-linear setting (Bocquet, 2011). We parameterize the surrogate modelling function $\widetilde{\mathcal{M}}(\boldsymbol{x}_t)$ as an additive model, where the statistical model $g_{\boldsymbol{\theta}}(\boldsymbol{x}_t)$ with its parameters $\boldsymbol{\theta}$ represents the residual,

$$\boldsymbol{x}_{t+\Delta t} \approx \widetilde{\mathcal{M}}(\boldsymbol{x}_t) = \boldsymbol{x}_t + g_{\boldsymbol{\theta}}(\boldsymbol{x}_t). \tag{20}$$

In our transfer learning experiments, the statistical model works on top of features extracted by the pre-trained network $\boldsymbol{\phi}(\boldsymbol{z}_\tau, \tau)$, as explained in Sect. 3.1. We fix the pseudo-time in the feature extractor and concatenate features from different pseudo-time steps. We have three different pseudo-time step settings, either a single step $\tau = [400]$, two pseudo-time steps $\tau = [50, 400]$, or six steps $\tau = [0, 50, 100, 200, 400, 600]$. On top of the feature extractor, we either train a linear regression or a

shallow neural network,

$$g_{\boldsymbol{\theta}}(\boldsymbol{x}_t) = \begin{cases} \mathbf{W}^\top \boldsymbol{\phi}(\boldsymbol{x}_t, \tau) + \boldsymbol{\beta} & \text{linear regression} \\ \mathbf{W}_1^\top \max\left(0, \mathbf{W}^\top \boldsymbol{\phi}(\boldsymbol{x}_t, \tau) + \boldsymbol{\beta}\right) + \boldsymbol{\beta}_1 & \text{shallow neural network,} \end{cases} \tag{21}$$

with its transposed weights $\mathbf{W}^\top$, $\mathbf{W}_1^\top$ and biases $\boldsymbol{\beta}, \boldsymbol{\beta}_1$. The shallow neural network has always 256 hidden features.

We compare the transfer learned surrogate models to random Fourier features (RFF, Rahimi and Recht, 2008) and neural networks trained from scratch. In the case of RFFs, we replace the pre-trained feature extractor by either 256 or 1536 random

Fourier features that approximate a Gaussian kernel as specified in Eq. (1) of Sutherland and Schneider (2015). These features can be seen as non-recurrent instantiation of a random feature extractor, often used for machine learning in dynamical systems (e.g., Vlachas et al., 2020; Arcomano et al., 2020). For the neural network, we use two different architectures, a simple architecture where we stack $m \in [1, 2, 3]$ fully-connected layers with 256 neurons and rectified linear unit (relu) activation functions in-between or a ResNet-like architecture with 3 residual blocks, as similarly specified in Appendix D1.

For all experiments, we optimize the statistical model for the mean-squared error (MSE) between the true increment $\Delta \boldsymbol{x}_{t+\Delta t} = \boldsymbol{x}_{t+\Delta t} - \boldsymbol{x}_t$ and the output of the neural network. The linear regression models are analytically estimated as $L_2$-regularized least-squares solution, also called ridge regression. The $L_2$-regularization parameter is hold constant ($\lambda = 1 \times 10^{-4}$) to simplify the training and the comparison between experiments. All neural network models are trained with the Adam optimizer ($\gamma = 5 \times 10^{-5}$) for 100 epochs with a batch size of 16384. We refrain from learning rate scheduling or early stopping for

the ease of comparison.




To evaluate the surrogate models and their stability for longer lead times than $\Delta t = 0.1\,\mathrm{MTU}$, we use the forecast as initial conditions for the next iteration, iterating for $k$ times to cover a lead time of $k \cdot \Delta t$. These trajectories are compared to the trajectories in the testing dataset in terms of root-mean-squared error, normalized by the state climatology in the training dataset. Additionally, to estimate the quality of the resulting climatology, we compare the Hellinger distance of the predictions between $5\,\mathrm{MTU}$ and $10\,\mathrm{MTU}$ to the testing dataset. We estimate the Hellinger distance as in Sect. 4.1.

### 4.2.1 Results

In Table 3, we evaluate the transfer learned surrogate models for a lead time of $\Delta t = 0.1\,\mathrm{MTU}$ and $\Delta t = 1\,\mathrm{MTU}$, which corresponds to 1 iterations or 10 iterations, respectively, and the Hellinger distance. Additionally, we compare this model to other surrogate models, learned from scratch.

**Table 3.** Performance of the surrogate models as average and standard deviation across ten random seeds. Shown is the normalized root-mean-squared error (nRMSE) for $\Delta t = 0.1\,\mathrm{MTU}$, as trained for, and after $\Delta t = 1.0\,\mathrm{MTU}$, and the Hellinger distance ($H$) of the prediction to the testing dataset. For visual purpose, the scores are multiplied by the number in the brackets. Params is the number of trainable parameters for the specified experiment. Downward arrows show the lower the score, the better. Bold values indicate the lowest scores for a given column.

| Model | Params | $\Delta t = 0.1$ nRMSE ($\times 10^5$) ↓ | $\Delta t = 1.0$ nRMSE ($\times 10^2$) ↓ | $H$ ($\times 10^2$) ↓ |
|---|---|---|---|---|
| RFF (256, linear) | $7.7 \times 10^2$ | $1603.9_{\pm 508.5}$ | $519.3_{\pm 83.6}$ | $92.1_{\pm 5.5}$ |
| RFF (1536, linear) | $4.6 \times 10^3$ | $17.4_{\pm 9.1}$ | $8.2_{\pm 50.5}$ | $6.7_{\pm 13.1}$ |
| RFF (256, NN) | $6.6 \times 10^4$ | $26.3_{\pm 2.7}$ | $71.4_{\pm 18.4}$ | $47.6_{\pm 13.3}$ |
| RFF (1536, NN) | $3.9 \times 10^5$ | $11.5_{\pm 1.6}$ | $29.1_{\pm 6.0}$ | $18.8_{\pm 5.2}$ |
| Dense ($\times 1$) | $1.8 \times 10^3$ | $17.2_{\pm 0.8}$ | $11.1_{\pm 0.7}$ | $3.1_{\pm 0.3}$ |
| Dense ($\times 2$) | $6.8 \times 10^4$ | $8.9_{\pm 1.4}$ | $8.5_{\pm 2.9}$ | $1.8_{\pm 0.1}$ |
| Dense ($\times 3$) | $1.3 \times 10^5$ | $10.2_{\pm 1.5}$ | $9.9_{\pm 3.5}$ | $\mathbf{1.7_{\pm 0.2}}$ |
| ResNet | $4.0 \times 10^5$ | $\mathbf{7.5_{\pm 2.3}}$ | $8.4_{\pm 4.8}$ | $\mathbf{1.7_{\pm 0.2}}$ |
| Transfer (Untrained, $6 \times \tau$, linear) | $4.6 \times 10^3$ | $5710.6_{\pm 0.0}$ | $100.1_{\pm 0.0}$ | $100.0_{\pm 0.0}$ |
| Transfer ($\tau = 400$, linear) | $7.7 \times 10^2$ | $115.7_{\pm 4.7}$ | $35.1_{\pm 1.5}$ | $18.6_{\pm 1.7}$ |
| Transfer ($6 \times \tau$, linear) | $4.6 \times 10^3$ | $23.1_{\pm 1.2}$ | $15.9_{\pm 1.8}$ | $6.7_{\pm 6.2}$ |
| Transfer ($\tau = 400$, NN) | $6.6 \times 10^4$ | $9.1_{\pm 1.2}$ | $8.4_{\pm 2.1}$ | $2.0_{\pm 0.2}$ |
| Transfer ($2 \times \tau$, NN) | $1.3 \times 10^5$ | $\mathbf{7.5_{\pm 2.0}}$ | $\mathbf{6.8_{\pm 1.0}}$ | $\mathbf{1.7_{\pm 0.2}}$ |
| Transfer ($6 \times \tau$, NN) | $3.9 \times 10^5$ | $12.4_{\pm 2.8}$ | $13.9_{\pm 7.9}$ | $14.1_{\pm 8.3}$ |

All transfer learned models have a predictive power that reaches beyond one model time unit. The performance of the pre-trained models is unreachable for untrained feature extractors, showing the added value of pre-training as DDMs. Extracting



features at multiple pseudo-time steps strengthens transfer learning with a linear regression compared to the models with only $\tau = 400$ as pseudo-time step for the feature extraction.

Using a shallow neural network with 256 hidden neurons after the feature extraction, only small differences between the transfer learned model with a single or multiple pseudo-time step are left. An increasing number of extracted features results into an increased collinearity between features and a more unstable training as shown by the increased standard deviation in the case of the *Transfer ($6 \times \tau$, NN)* experiment. The best performing transfer learned model is the one with features at two pseudo-time steps and neural network as last layer, *Transfer ($2 \times \tau$, NN)*, with $\tau = [50, 400]$. This model balances additional information from more features with increased collinearity.

The $L_2$-regularization of the linear regression reduces the feature collinearity, such that *Transfer ($6 \times \tau$, linear)* performs better than the *Transfer ($\tau = 400$, linear)* model. Consequently, features from multiple pseudo-time steps increase the performance for the linear regression case, whereas fine-tuning with a neural network can lead to better results, if taken care of the collinearity.

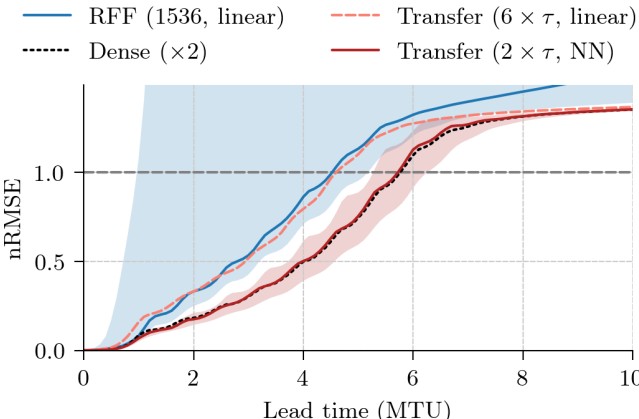

**Figure 7.** The normalized root-mean-squared error (nRMSE) as function of integration time steps for random Fourier features (RFF) with 1536 features and a linear regression, a *dense* neural network with two layers trained from scratch, and transfer learned models (Transfer) with features from six tipseudo-time steps with a linear regression and from two pseudo-time steps with a neural network. Shown is the median across ten different random seeds. Additionally, for the RFF (1536, linear) and the Transfer ($2 \times \tau$, NN) experiments, the 5th and 95th percentile is depicted as shading.

The transfer learned model with features from a single pseudo-time steps, *Transfer ($\tau = 400$, linear)*, performs better than random Fourier features with the same number of features, *RFF (256, linear)*. Using the same number of features as for the multistep model, random Fourier features have a lower initial error than the transfer learned model. However, if the surrogate model is cycled, the performance and stability of the random Fourier features heavily depend on the drawn random weights, whereas the transfer learned model is stable for all tested integration time steps, as also shown in Fig. 7. The transfer learned models converges towards a climatological forecast, as can be also seen in the performance of the *Transfer (Untrained, $6 \times \tau$, linear)* model, whereas the random Fourier based models diverge. Additionally, as random features scale with the data



dimensionality, transfer learning can be preferable for higher-dimensional problems than the Lorenz 1963 system. Therefore, transfer learning can outperform random Fourier features, especially with regard to the long-term stability of the model.

The transfer learned model with the linear regression is similar to the performance of a shallow neural network, and the transfer learned model with the neural network can perform better than the tested neural network architectures that were trained 415 from scratch. However, the results for the neural networks trained from scratch may indicate convergence issues because of the fixed learning rate. In total, the results indicate that transfer learned models can perform similarly or better than deep neural networks trained from scratch and random Fourier features.

To see if a better generative score translates into better surrogate models, we compare the generative Hellinger distance to the RMSE of the surrogate model, transfer learned with a linear regression for different output parameterizations in Tab. 4. 420 In general, the ordering between the output parameterizations remains more or less the same for surrogate modelling as for data generation; the state parameterization has the worst performance, whereas the noise and velocity parameterization have a performance similar to each other. For the noise and velocity parameterization pre-training with a cosine noise scheduling performs better than with a linear scheduling.

**Table 4.** Comparison between the pre-trained denoising diffusion models in terms of generative Hellinger distance $H$ and root-mean-squared error (nRMSE) after one iteration with the transfer learned surrogate model ($\Delta t = 0.1\,\mathrm{MTU}$). The brackets in the experiment name indicate the noise scheduling with which the denoising diffusion model was trained. Shown is the averaged performance and standard deviation across ten different seeds. For visual purpose, the scores are multiplied by the number in the brackets. Downward arrows show the lower the score, the better. The experiment in bold is the experiment used in Tab. 3 and bold numbers indicate the best performance in a given column.

| | Generative | Surrogate ($\Delta t = 0.1$) |
|---|---|---|
| Experiment | $H$ ($\times 10^2$) $\downarrow$ | nRMSE ($\times 10^3$) $\downarrow$ |
| State $\boldsymbol{x}$ (linear) | $17.61_{\pm 2.24}$ | $4.13_{\pm 0.31}$ |
| State $\boldsymbol{x}$ (cosine) | $14.38_{\pm 1.69}$ | $4.31_{\pm 0.22}$ |
| Noise $\boldsymbol{\epsilon}$ (linear) | $4.56_{\pm 0.30}$ | $3.83_{\pm 0.26}$ |
| Noise $\boldsymbol{\epsilon}$ (cosine) | $\mathbf{4.02}_{\pm \mathbf{0.19}}$ | $2.74_{\pm 0.13}$ |
| Velocity $\boldsymbol{v}$ (linear) | $4.56_{\pm 0.32}$ | $3.08_{\pm 0.88}$ |
| **Velocity $\boldsymbol{v}$ (cosine)** | $\mathbf{3.96}_{\pm \mathbf{0.16}}$ | $\mathbf{2.34}_{\pm \mathbf{0.12}}$ |

The differences caused by different random seeds is smaller than the differences between different parameterizations and 425 noise scheduler; experiments with a cosine noise scheduler generally result into lower standard deviations between seeds. Therefore, the better the generative model, the better can the NN be fine-tuned towards surrogate modelling. Pre-training with a velocity parameterization hereby results into the best surrogate modelling performance.



## 4.3  Data assimilation with a generated ensemble

We test how the latent states in the DDM can be used for ensemble generation in a data assimilation setup. We define the
long trajectories from the testing dataset as our truth $x^t_{1:t_{end}}$, going from time $t = 1$ to time $t = t_{end}$. In our experiments, the
observation operator is given as identity matrix $\mathbf{H} = \mathbf{I}$, all three states are observed. The observations $y^o_t$ at time $t$ are perturbed
with white noise drawn from a Gaussian distribution with a pre-specified observation standard deviation $\sigma^o$,

$$y^o_t = \mathbf{H}x^t_t + \epsilon^o_t, \qquad\qquad\qquad \epsilon^o_t \sim \mathcal{N}(\mathbf{0}, (\sigma^o)^2\mathbf{I}). \qquad\qquad (22)$$

If not differently specified, we set the observation standard deviation to $\sigma^o = 2$ and the time interval between observations to
$\Delta t = 0.1\,\mathrm{MTU}$.

As data assimilation algorithm, we use an ensemble transform Kalman filter (ETKF, Bishop et al., 2001; Hunt et al., 2007).
In cases where the ensemble is externally generated, we modify the ETKF as proposed by Schraff et al. (2016) to update the
deterministic background forecast.

In the DDM experiment, based on a deterministic run, we draw 50 ensemble members with the DDM as proposed in Sect.
3.2. We use the pre-trained neural network with a velocity output parameterization, a cosine noise schedule, and a single random
seed (seed = 0) to generate the ensemble. The denoising diffusion model is additionally defined by its sampling scheme, the
number of maximum pseudo-time steps $T$ until the prior distribution is reached, and the signal magnitude $\alpha_\tau$ of the partial
diffusion. We sample in the denoising process with a DDPM scheme and set the number of maximum pseudo-time steps
to $T = 100$, reducing the computational needs. The only parameter in the ensemble generation is consequently the signal
magnitude $\alpha_\tau$, which we tune for each experiment independently.

We compare the proposed ensemble generation methodology to a full ETKF, ensemble optimal interpolation (EnOI), and
3D-Var. The full ETKF is the reference and includes flow-dependent covariances, a feature missing in the proposed ensemble
generation method. The EnOI experiments define a baseline with static covariance matrices. As we have to generate an external
ensemble in the EnOI experiments, we induce sampling errors in the data assimilation. The 3D-Var experiments use the same
covariances as the EnOI but analytically solve the Kalman filter equation without sampling.

We run the full ETKF with 3 or 12 ensemble members and an optimally tuned multiplicative prior covariance inflation. Here,
the ensemble mean specifies the deterministic forecast. The ETKF estimates per forecast one first guess covariance matrix $\mathbf{P}^b_t$.
To define the background covariances in our EnOI experiments, we use the first guess covariance matrices from the 12 member
ETKF run with an update time delta of $\Delta t = 0.1\,\mathrm{MTU}$, averaged over the full trajectory with $S$ steps and inflated by tuning
factor $\alpha$,

$$\mathbf{B} = \begin{cases} \alpha S^{-1}\sum_{t=1}^{S}\mathbf{P}^b_t & \text{full covariance} \\ \alpha S^{-1}\sum_{t=1}^{S}(\sigma^b_t)^2\mathbf{I} & \text{diagonal covariance.} \end{cases} \qquad\qquad (23)$$

The full covariance matrix is specified in Appendix D2. Besides a full covariance, we also specify a diagonal covariance,
an often used approximation in EnOI. With the so defined background covariances we draw per update step 50 ensemble





members (for a fair comparison to the DDM experiments), centered around the deterministic run, and then used to update the

deterministic run.

We initialize all experiments with states randomly drawn from the climatology of the testing dataset. Each experiment has 55000 update cycles, where we omit the first 5000 updates as burn-in phase. We repeat each experiment 16 times with different random seeds. This way, each batch of experiments has $8 \times 10^5$ analyses. The parameters for each batch of experiments, the signal amplitude $\alpha_\tau$ for the partial diffusion in the DDPM scheme, the prior covariance inflation $\rho$ for the ETKF, and

the covariance inflation $\alpha$ for the EnOI, are tuned to give the lowest time-averaged analysis RMSE, averaged over the 16 repetitions.

We compare the analyses to the true trajectories in terms of RMSE at analysis time, normalized with respect to the climatology. Additionally, we run forecasts based on the analyses and compare them to the true trajectories to see the impact on longer lead times.

### 4.3.1  Results

We compare the analysis and background normalized RMSE to a tuned ensemble Kalman filter (ETKF), ensemble optimal interpolation (EnOI), and 3D-Var in Table 5. The time between updates is $\Delta t = 0.1\,\mathrm{MTU}$, a mildly non-linear case (Bocquet, 2011).

**Table 5.** Comparison of the normalized root-mean-squared error (nRMSE) in the analysis and background forecasts for different data assimilation methods. Shown are the nRMSE average and standard deviation across 16 experiments with different seeds. The number in the brackets is the number of ensemble members, *Diag* the use of a diagonal covariance matrix, and *Full* the use of a full covariance matrix.

| Experiment | Analysis | Background |
|---|---|---|
| EnOI Diag ($\times 50$) | $0.158_{\pm 0.001}$ | $0.209_{\pm 0.003}$ |
| EnOI Full ($\times 50$) | $0.138_{\pm 0.001}$ | $0.189_{\pm 0.001}$ |
| 3D-Var | $0.136_{\pm 0.001}$ | $0.189_{\pm 0.002}$ |
| ETKF ($\times 3$) | $0.089_{\pm 0.003}$ | $0.130_{\pm 0.003}$ |
| ETKF ($\times 12$) | $0.077_{\pm 0.003}$ | $0.114_{\pm 0.004}$ |
| DDM ($\times 50$) | $0.135_{\pm 0.001}$ | $0.188_{\pm 0.002}$ |

The ensemble members generated by a DDM can be used to assimilate observations into a deterministic forecast with a

EnOI-like scheme. Data assimilation with DDM results into a slight improvement compared to EnOI with full covariances and matches 3D-Var; it hereby surpasses the performance of EnOI with a diagonal covariance, as often employed in EnOI. As the DDM has just one tuning parameter, the signal magnitude $\alpha_\tau$, this type of ensemble generation can provide a simplified way for ensemble data assimilation algorithms, needing less tuning than EnOI.

The DDM is state-dependent, which can explain the small advantage compared to EnOI with the same number of ensemble

members. However, as an ensemble Kalman filter provides additionally a flow dependency, it performs much better than EnOI,




3D-Var, and the DDM. Consequently, to match the performance of the ensemble Kalman filter with the DDM, we need to incorporate flow information into the DDM. Nevertheless, pre-trained for state generation, DDMs can be a cheap alternative to generate an ensemble for data assimilation purposes.

In Fig. 8, we show the influence of the signal magnitude on the background RMSE and the spread of the generated ensemble.

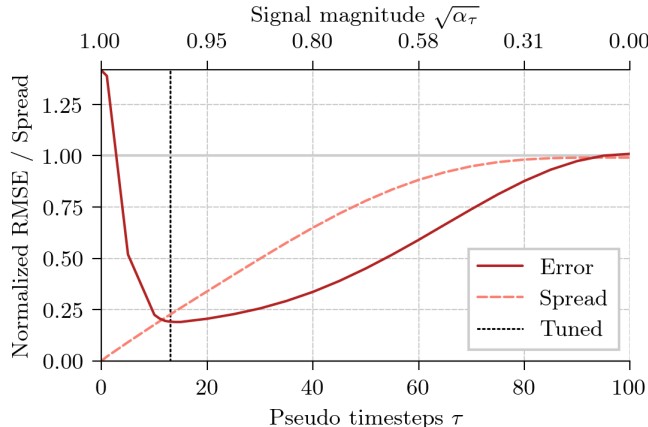

**Figure 8.** Normalized root-mean-squared error (red solid line) of the deterministic background forecast and generated ensemble standard deviation (orange dashed line). The 50 ensemble members of the EnOI experiments are generated with the diffusion sampler with different signal magnitudes. The black dotted line corresponds to the signal magnitude used for the DDM experiments in Table 5. The results are averaged across 16 experiments.

The bigger the pseudo-time step, the smaller the signal magnitude, and the more diffused is the deterministic run during the partial diffusion, which controls the degree of uncertainty in the ensemble. For a very small pseudo-time step with a signal magnitude near one, $\alpha_\tau \approx 1$, almost no noise would be added, and we would end up with a (too) small ensemble spread. For a large pseudo-time step with a signal magnitude near zero, $\alpha_\tau \approx 0$, almost all data would be replaced by noise in the latent state; the generated ensemble would correspond to a climatological ensemble with a (too) large ensemble spread. Similar to 490    an ensemble data assimilation system, the lowest RMSE is reached when the ensemble spread roughly matches the RMSE. Consequently, the choice of the pseudo-time step is similar to the covariance inflation factor in an ensemble data assimilation system.

Until now, we have shown results for a single time delta between two update times with $\Delta t = 0.1 \, \text{MTU}$. In Fig. 9, we show results for varying this time delta for the ETKF, the EnOI with static covariances, and EnOI with the DDM sampler; all 495    experiments are tuned for each time delta independently.

Increasing the time between data assimilation updates increases the non-linearity of the system. For all tested time deltas, EnOI with a DDM performs in the analysis RMSE slightly better than EnOI with static covariances. Although the performance of the ETKF is unreachable for any update time, the gain of DDMs compared to static covariances increases with increasing non-linearity of the system.



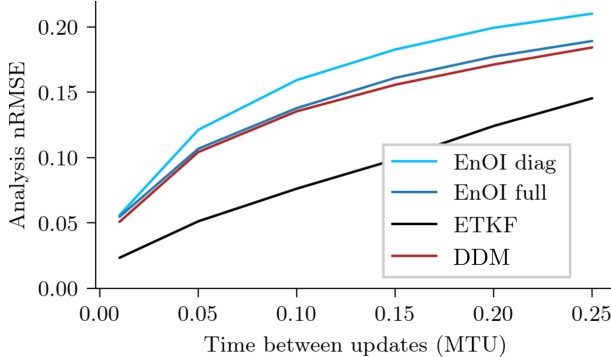

**Figure 9.** Scaling of the analysis nRMSE with increasing time delta between updates for the ETKF, EnOI with drawn ensemble members from a climatological covariance, EnOI with an exact climatological covariance, and EnOI with ensemble members generated by a denoising diffusion model. The parameters of the different data assimilation methods are tuned to reduce the nRMSE.

This gain is a result of the non-Gaussian distribution of the generated ensemble members for heavily non-linear state propagation, as can be seen in Fig. 10. The latent state obtained after the diffusion process is purely Gaussian distributed by definition, see also Eq. (4). However, the iterative denoising process is state-dependent, which can result into non-Gaussian ensemble distributions. The larger the pseudo-time step for the ensemble generation, the larger the sampled portion of the attractor and the more non-Gaussian can get the distribution in the denoising process. Therefore, tuning the pseudo-time step of the DDM

allows us not only to tune its ensemble spread but also the sampled portion of the attractor.

## 5   Summary and discussion

In this study, we investigate unconditional denoising diffusion models (DDMs) for representation learning in dynamical systems. We train such models on the task of state generation in the Lorenz 1963 model. Using a large dataset of states and deep residual neural networks (NNs), we test settings nearly unlimited from the sample and NN size. In these settings, the DDM can

generate states that are almost indistinguishable to states drawn from the model's attractor.

    Our results for state generation correspond to those found for image generation. Predicting the noise performs better than directly predicting the state, as similarly found by Ho et al. (2020). Across all tested settings, a cosine noise scheduling is superior to a linear noise scheduling (Nichol and Dhariwal, 2021). Furthermore, we obtain the most stable results for predicting a velocity $v$, as discussed by Salimans and Ho (2022). For a few generation steps, using deterministic sampling with denoising

diffusion implicit models (DDIM) outperforms its stochastic counterparts (Song et al., 2020a). In general, results from image generation and improvements therein seem transferable to state generation for dynamical systems.

    We can approximate the state distribution of the Lorenz 1963 system by state quantization and estimating a three-dimensional empirical probability density function (PDF). We can consequently evaluate generative models by comparing the empirical



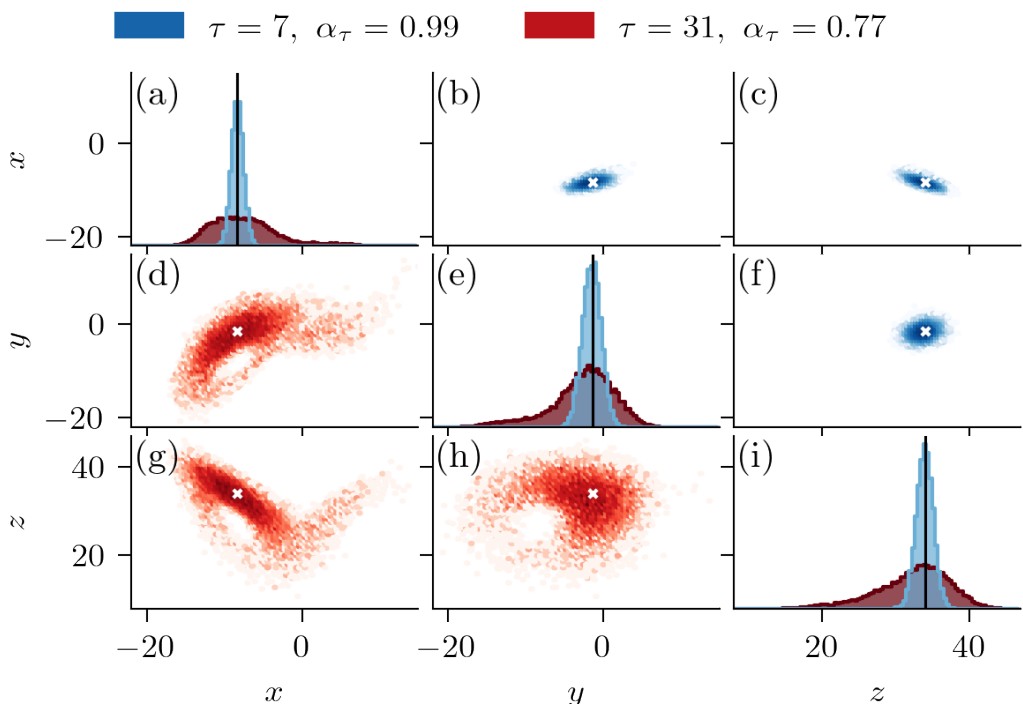

**Figure 10.** Comparison of two ensembles with $16384$ members, generated by a denoising diffusion model with a DDPM sampler for $\tau = 7$ (blue) and $\tau = 31$ (red) pseudo-time steps based on an arbitrary deterministic state. These number of pseudo-time steps is used in Fig. 9 for $\Delta t = 0.05$ and $\Delta t = 0.25$, respectively. The diagonal panels show the empirical probability density function of the generated ensemble members in (a) $x$-direction, (e) $y$-direction, and (i) $z$-direction. The inter-variable relationships as two-dimensional projection are shown in the off-diagonal panels, with the upper triangular panels (b, c, f) for $\tau = 7$ and the lower triangular panels (d, g, h) for $\tau = 31$. The black lines and white stars are the deterministic state.

PDF of states generated with the model to states drawn from the attractor. However, the estimation of a multivariate empirical
PDF is infeasible for higher-dimensional systems.

To circumvent such problems, we adapt the Fréchet inception distance (FID) to geoscientific settings. This distance compares the generated states to the real states with the Fréchet distance in the feature space of a pre-trained NN. We replace the commonly used inception network (Szegedy et al., 2014) by a NN pre-trained for surrogate modelling in the Lorenz 1963 system. The ordering of different generative parameterizations in this adapted metric is very similar to the ordering found by
estimating the Hellinger distance between the empirical PDFs. An adapted FID can be a good metric for evaluating generative models in high-dimensional geoscientific settings. One of the remaining challenges hereby is the choice of an appropriate pre-trained NN to estimate the Fréchet distance.





At a first glance, unconditional DDMs, trained for state generation, have a smaller application range compared to their conditional counterpart. Here, we fine-tune the unconditional denoising NN for surrogate modelling and apply the full unconditional
DDM for ensemble generation.

By removing its last layer, we can use the denoising NN as a feature extractor for surrogate modelling. Our results indicate that the NN learns general features about the dynamical system, when pre-trained for state generation. The extracted features depend on the pseudo-time step, with more complex features for smaller steps. Consequently, by combining features from different pseudo-time steps, we use more information from the feature extractor.

Although the DDM has previously never seen information about the temporal dynamics of the dynamical system, we can fine-tune the denoising NN for surrogate modelling. By regressing features from a single pseudo-time step, the fine-tuned network performs better than random Fourier features with the same number of extracted features. This suggests that for higher-dimensional problems, pre-trained denoising NNs may perform much better than random Fourier features as feature extractor.

By learning a shallow NN on top of the extracted features, the fine-tuned network achieves scores similar to deep NNs. As Lorenz 1963 is a low-dimensional system, where NNs can almost perfectly predict the temporal dynamics, a NN trained from scratch can be difficult to beat. However, for high-dimensional systems, where we might have only a few training samples, training a deep NN from scratch might be infeasible. To pre-train a DDM, we can use large, heterogeneous, datasets and, then, fine-tune the NN on small problem-specific datasets. Our encouraging results for low-dimensional settings indicate this
potential for transfer learning of DDMs and their use as pre-trained feature extractor.

Beside surrogate modelling, we apply the DDM for ensemble generation in a data assimilation. By diffusing and denoising, we generate an ensemble out of a deterministic run. The ensemble can define the prior distribution for an ensemble optimal interpolation (EnOI) scheme to assimilate observations into a deterministic forecast. Such a data assimilation with a DDM as sampler performs at least as good as EnOI with static but tuned covariances. The ensemble generated with the DDM inherits
the state-dependency of the denoising NN. As a result, the more non-linear the system, the larger can get the gain of the DDM sampling compared to static covariances in EnOI.

Data assimilation with a propagated ensemble profits from its state- and flow-dependency. Since the DDM is only trained for state generation, its attractor is only defined in state space. The time dimension and, hence, one of the requirements for flow dependency is missing. Consequently, the performance of a tuned ensemble Kalman filter is unreachable for EnOI with
555 a DDM sampler. To make the sampler flow dependent, we must incorporate the time dimension. Instead of generating states static in time, the DDM could learn to generate small trajectories. By partially diffusing the forecast trajectory, we can inform the sampler about the temporal development and, thus, generate a flow-dependent ensemble.

The proposed ensemble sampling could additionally augment an ensemble by additional ensemble members. This augmentation would be similar to the use of a climatological augmented ensemble (Kretschmer et al., 2015). The climatological
ensemble members would be replaced by ensemble members generated with the DDM. This can be additionally seen like ensemble data assimilation with a hybrid covariance matrix (Hamill and Snyder, 2000; Lorenc, 2003), where the covariance is a weighted average between the original ensemble covariance and the covariance of the generated members. On the one hand,





the original ensemble members would bring the flow-dependency into ensemble. On the other hand, augmenting the ensemble by generated members could be a way to reduce the need of inflation and localization in an ensemble data assimilation system.

The application of pre-trained unconditional DDMs for surrogate modelling and ensemble generation indicate their potential for geoscientific problems. Trained to sample from the attractor, the model learns an internal representation, then applicable in downstream tasks. The combination of DDMs with data assimilation could additionally be a way to learn such deep generative models from combining observations with a geophysical model. Using such a combination, DDMs could possibly learn a representation of the true Earth system's attractor. This representation might be then helpful for large-scale applications like

model error corrections (e.g. Bonavita and Laloyaux, 2020; Farchi et al., 2021; Chen et al., 2022; Finn et al., 2023) or digital twins (e.g. Bauer et al., 2021a, b; Latif, 2022; Li et al., 2023).

## 6    Conclusions

In this manuscript, we investigate the capabilities of denoising diffusion models for representation learning in dynamical systems. Based on our results with the Lorenz 1963 model, we conclude the following:

– Denoising diffusion models can be trained to generate states on the attractor of the dynamical system. Using a large training dataset and a residual neural network, the generated states are almost indistinguishable to states drawn from the true attractor. To achieve a stable training for dynamical systems, we can recommend denoising diffusion models with a velocity $v$ output parameterization and a cosine noise scheduler. Similar to results for image generation, the deterministic DDIM sampling scheme works best for few pseudo-time steps.

– Denoising diffusion models can be fine-tuned for downstream tasks by applying the denoising neural network as feature extractor and retraining its last layer. The features extracted by the denoising network depend on the used pseudo-time step with more complex features for smaller steps. Combining features at different pseudo-time steps, we can empower the feature extractor for downstream tasks. A better performing generative model can hereby also achieve better scores in downstream tasks.

– Pre-trained as denoising diffusion model for state generation, neural networks can be transfer learned for surrogate modelling. Their performance in these low-dimensional settings is similar to the deep neural network trained from scratch. Training neural networks as denoising diffusion models has therefore the potential for large-scale pre-training of deep neural networks for geoscientific problems.

– The pre-trained denoising diffusion model can be applied to generate an ensemble out of a deterministic run. By partial

diffusion and denoising with the neural network, we can sample from the attractor in the run's surrounding. As tuning parameter, we can choose the number of diffusion steps, which controls the portion of the sampled attractor and the resulting ensemble spread. Since the denoising network is trained for static state generation, the generated ensemble is state-dependent but lacks flow dependency. To introduce such a flow dependency, the denoising diffusion model must be also trained with time-dependent states, e.g., by training to generate trajectories.



– The ensemble generated with a pre-trained denoising diffusion model can define the prior distribution for ensemble
optimal interpolation to assimilate observations into a deterministic forecast. Data assimilation with this sampler can
outperform ensemble optimal interpolation with tuned climatological covariances. The more non-linear the dynamical
system, the larger can get the gain of sampling a denoising diffusion model compared to static covariances.

## Appendix A:  A dynamical system point of view on denoising diffusion models

The diffusion process progressively replaces the signal by noise, and the denoising NN is trained to invert this process and can
be iteratively used to generate a signal out of pure noise. In our case, we train the NN to generate states that should lay on the
attractor of a dynamical system. The process of generate a signal out of pure noise resembles the spin-up procedure often used
for dynamical systems.

To spin-up dynamical systems, random fields are generated that lay in the basin of attraction for the dynamical system. If
these randomly sampled states are integrated with the dynamical system for enough steps, all states from the basin of attraction
converge to the attractor of the dynamical system.

In fact, the diffusion process corresponds to an Itô stochastic differential equations (SDE, Song et al., 2021), integrated in
pseudo-time,

$$dz = \mathrm{m}(z, \tau)dt + \mathrm{g}(\tau)dw, \tag{A1}$$

where $\mathrm{m}(z, \tau)$ is the drift coefficient, $\mathrm{g}(\tau)$ the diffusion coefficient, $dt$ an infinitesimal-small pseudo-time step, and $dw$ defines
a Wiener (Brownian) process. We can use ancestral sampling to integrate this SDE, going from the state distribution $p(x) =
p(z_0)$ at $t = 0$ to the prior distribution $p(z_T)$, Eq. (5), at $t = T$. Defining $\mathrm{m}(z, \tau) = -\frac{1}{2}\beta(\tau)z$ and $\mathrm{g}(\tau) = \sqrt{\beta(\tau)}$ with noise
scales $\beta(\tau)$, the *variance-preserving* diffusion process is recovered (Song et al., 2021).

Inverting the process, we start at pseudo-time $t = T$ with pure noise, drawn from the prior distribution $p(z_T)$, and move
towards the state distribution target $p(z_0)$. The inverse of a diffusion process is again a diffusion process, resulting into the
reverse-time SDE,

$$dz = \left[\mathrm{m}(z, \tau)dt - \mathrm{g}(\tau)^2 \nabla_z \log p(z)\right] dt + \mathrm{g}(\tau)d\widetilde{w}, \tag{A2}$$

with $d\widetilde{w}$ as reverse-time Wiener process. Remarkably, the reverse-time SDE, Eq. (A2), is defined by the known drift and
diffusion coefficients and the so-called score function $\nabla_z \log p(z)$, the gradient that points towards the data-generating distri-
bution. Consequently, once the score function is known for all pseudo-time steps, we can use ancestral sampling to integrate
the denoising process, Eq. (A2), and to generate samples based on noise drawn from the pre-defined prior distribution.

Similarly to predicting the noise, the state, or an angular velocity, the score can be approximated by a NN, $s_\theta(z_\tau, \tau) \approx
\nabla_{z_\tau} \log p(z_\tau)$. Predicting the noise is hereby proportional to the approximated score function by the relation

$$\widehat{\epsilon}_\theta(z_\tau, \tau) = -\sigma_\tau s(z_\tau, \tau), \tag{A3}$$



the predicted noise points away from data-generating distribution. Consequently, training the NN to predict the noise, Eq. (13), is equivalent with score matching (Hyvärinen, 2005; Vincent, 2011; Song et al., 2020b).

The simplest method to integrate the SDEs is using an Euler-Maruyama integration with a fixed step size. This leads to similar procedures as specified in Sect. 2. Consequently, sampling with the DDPM or DDIM scheme, as specified by Eq. (14) and Eq. (15a), corresponds to special discretizations of the reverse-time SDE (Song et al., 2020a, 2021), defined in Eq. (A2).

However, the formulation of the diffusion process as SDE allows us to use different integration methods with adaptive step sizes (Jolicoeur-Martineau et al., 2022; Dockhorn et al., 2022; Lu et al., 2022) such that fewer integration steps than with the DDPM scheme are needed to generate data.

The generation of states with the SDE is a sort of dynamical system, integrated in pseudo-time. The smaller the integration error, the smaller the approximation error, and the larger the number of training samples, the smaller the error between the

distribution of generated states and the data-generating distribution (De Bortoli, 2022), with convergence in its limit. Therefore, if the NN is trained on samples that lay on the attractor of the dynamical system, the generated samples will also lay on this attractor in these limits.

## Appendix B: Data assimilation prior distributions from denoising diffusion models

In the Bayesian formalism of data assimilation, we want to find the posterior distribution $p(\boldsymbol{x}_t \mid \boldsymbol{y}_{1:t})$ of the current state $\boldsymbol{x}_t$ at

time $t$ based on all observations $\boldsymbol{y}_{1:t}$ up to the very same time. We can use Bayes' theorem to split the posterior distribution into a prior distribution $p(\boldsymbol{x}_t \mid \boldsymbol{y}_{1:t-1})$ and the observation likelihood $p(\boldsymbol{y}_t \mid \boldsymbol{x}_t)$,

$$p(\boldsymbol{x}_t \mid \boldsymbol{y}_{1:t}) = \frac{p(\boldsymbol{y}_t \mid \boldsymbol{x}_t)\,p(\boldsymbol{x}_t \mid \boldsymbol{y}_{1:t-1})}{\int p(\boldsymbol{y}_t \mid \boldsymbol{x}')\,p(\boldsymbol{x}' \mid \boldsymbol{y}_{1:t-1})\,d\boldsymbol{x}'}. \tag{B1}$$

The influence of past observations $\boldsymbol{y}_{1:t-1}$ on the current state is hence encoded into the prior distribution. In data assimilation, we often estimate a deterministic analysis $\boldsymbol{x}_t^{\mathrm{a}}$ as single best estimate of the posterior. Having access to a geophysical forecast

model $\mathcal{M}(\cdot)$, we can generate a deterministic model forecast $\boldsymbol{x}_t^{\mathrm{f}}$ based on the analysis from the previous time $\boldsymbol{x}_t^{\mathrm{f}} = \mathcal{M}(\boldsymbol{x}_{t-1}^{\mathrm{a}})$. Using such a model forecast as basis, we can generate an ensemble with a denoising diffusion model as introduced in Sec. 3.2. This generated ensemble can be seen as specifying the prior distribution in the Bayes' theorem, Eq. (B1),

$$p(\boldsymbol{x}_t \mid \boldsymbol{y}_{1:t-1}) = \int p(\boldsymbol{x}_t \mid \boldsymbol{z}_{\tau,t}) p(\boldsymbol{z}_{\tau,t} \mid \boldsymbol{y}_{1:t-1})\,d\boldsymbol{z}_{\tau,t} \tag{B2}$$

$$= \int \int p(\boldsymbol{x}_t \mid \boldsymbol{z}_{\tau,t}) p(\boldsymbol{z}_{\tau,t} \mid \boldsymbol{x}_t^{\mathrm{f}}) p(\boldsymbol{x}_t^{\mathrm{f}} \mid \boldsymbol{y}_{1:t-1})\,d\boldsymbol{x}_t^{\mathrm{f}} d\boldsymbol{z}_{\tau,t} \tag{B3}$$

$$= \int p(\boldsymbol{x}_t \mid \boldsymbol{z}_{\tau,t}) p(\boldsymbol{z}_{\tau,t} \mid \boldsymbol{x}_t^{\mathrm{f}})\,d\boldsymbol{z}_{\tau,t}, \qquad\qquad p(\boldsymbol{x}_t^{\mathrm{f}} \mid \boldsymbol{y}_{1:t-1}) = \delta\left[\boldsymbol{x}_t^{\mathrm{f}} - \mathcal{M}(\boldsymbol{x}_{t-1}^{\mathrm{a}})\right], \tag{B4}$$

$$= \int p(\boldsymbol{x}_t \mid \boldsymbol{z}_{\tau,t}) \mathcal{N}(\boldsymbol{z}_{\tau,t} \mid \sqrt{\alpha_\tau}\boldsymbol{x}_t^{\mathrm{f}}, (1-\alpha_\tau)\mathbf{I})\,d\boldsymbol{z}_{\tau,t} \tag{B5}$$

$$= \mathbb{E}_{\boldsymbol{z}_{\tau,t} \sim \mathcal{N}(\sqrt{\alpha_\tau}\boldsymbol{x}_t^{\mathrm{f}}, (1-\alpha_\tau)\mathbf{I})}[p(\boldsymbol{x}_t \mid \boldsymbol{z}_{\tau,t})]. \tag{B6}$$

The tuning factor for the prior distribution is the number of partial diffusion steps $\tau$/signal magnitude $\alpha_\tau$. We can sample from conditional state distribution given the diffused states $p(\boldsymbol{x}_t \mid \boldsymbol{z}_{\tau,t})$ by denoising the diffused states. By partially diffusing the




deterministic forecast for $\tau$ steps, sampling from the diffused distribution, and denoising with the neural network, we can thus specify the prior distribution for data assimilation.

**Appendix C: Noise scheduler**

We test two different noise scheduler, a linear scheduler and a cosine scheduler. The resulting signal and noise amplitude is shown in Fig. C1:

– The *linear scheduler* linearly increases the relative noise magnitude $\beta_\tau$ (Ho et al., 2020). The relative noise magnitude specifies then the relative signal magnitude $\alpha'_\tau = \beta_\tau$, used in Eq. (3). We linearly increase $\beta_\tau$ from 0.0001 for $\tau = 0$ to 0.03 for $\tau = T = 1000$. The signal and noise amplitude are shown as blue lines in Fig. C1.

   – The *cosine scheduler* defines the signal amplitude $\alpha_\tau = \cos(\frac{\pi}{2} \frac{\tau'+s}{1+s})$ as shifted cosine function with shift $s = 0.008$ and $\tau' = \frac{\tau}{1000}$ (Nichol and Dhariwal, 2021). This signal amplitude is directly used in Eq. 4, and shown alongside the noise

amplitude as red lines in Fig. C1.

The signal amplitude in the cosine scheduler decays slower than in the linear scheduler. Generating with a cosine scheduler consequently is more concentrated in a high signal-to-noise ratio regime, which results into a better state generation.

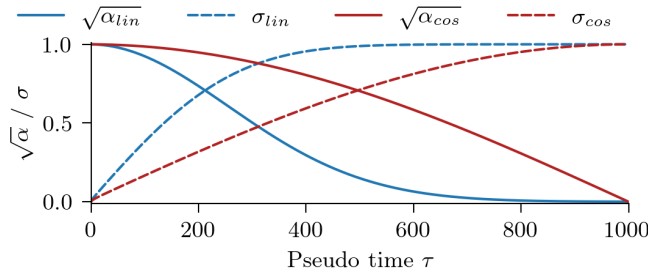

**Figure C1.** The signal (solid) and noise (dashed) factors for the here used linear noise scheduler (blue) and cosine noise scheduler (red).

**Appendix D: Configurations**

**D1  Denoising network**

As denoising neural network, we use a fully-connected residual neural network. We use a linear layer, mapping from the three-dimensional state vector to 256 features. Concurrently, we apply a sinusoidal time embedding (Vaswani et al., 2017), where we map the pseudo-time information into 128 sinusoidal features with increasing wave-lengths. After these initial mappings, we apply three residual blocks.





Each residual block has a branch. In the branch, the data is normalized by layer normalization. The affine transformations,
applied after the normalization, are linearly conditioned on the embedding (Perez et al., 2017). After normalizing and modu-
lating, we apply a shallow neural network with 256 features and a rectified linear unit (relu) as activation function. The output
of the branch is added to the input to the residual block.

After the three residual blocks, we linearly combine the extracted features to get the output of the neural network. Depending
on its parameterization, see also Sect. 2.4, the output has different meanings. In total, the denoising neural network has $1.2 \times 10^6$
parameters.

## D2  Background covariance matrix

The ensemble interpolation and 3D-Var experiments in Sect. 4.3 use a static background covariance matrix. The matrix is
based on averaged ensemble covariances from a tuned ensemble transform Kalman filter with 12 ensemble members. Scaled
by a tuning factor $\alpha$, the matrix is tuned for each experiment independently. The unscaled background covariance matrix is
specified in Table D1.

**Table D1.** The background covariance matrix as used in the ensemble optimal interpolation and 3D-Var experiments in Sect. 4.3.

|   | $x$ | $y$ | $z$ |
|---|---|---|---|
| $x$ | 0.520 | 0.747 | 0.000 |
| $y$ | 0.747 | 1.378 | 0.001 |
| $z$ | 0.000 | 0.001 | 1.177 |

## Appendix E:  Validation metrics

The scores are estimated based on two different distributions $q$, the generated sample distribution, and $p$, the testing sample
distribution. In total, we have five different metrics:

- The *Hellinger distance* is estimated between the empirical probability density functions of $q$ and $p$. For the estimation of
the Hellinger distance, we discretize the three-dimensional states into $k$ cubes (Scher and Messori, 2019) and count the
   probability for $q$ and $p$ that a sample lies in the $i$-th cube. In slight abuse of notation, we denote $q_i$ and $p_i$ as probabilities
   for $q$ and $p$, respecitvely.

$$H(q,p) = \frac{1}{\sqrt{2}}\sqrt{\sum_{i=1}^{k}(\sqrt{q_i} - \sqrt{p_i})^2}. \tag{E1}$$

   The distance should be $0$ if the distributions perfectly correspond.

- The *Fréchet surrogate distance* measures the distance between the distributions in a feature space $\varphi(\boldsymbol{x})$ of an independent
   neural network, here trained for surrogate modelling. For both distributions, the means, $\boldsymbol{\mu}_q$ and $\boldsymbol{\mu}_p$, and covariances, $\boldsymbol{\Sigma}_q$





and $\boldsymbol{\Sigma}_p$ in feature space are estimated. The Fréchet distance is then given as Wasserstein-2 distance with a multivariate Gaussian assumption.

$$\text{FSD}(\mathcal{N}(\boldsymbol{\mu}_q, \boldsymbol{\Sigma}_q), \mathcal{N}(\boldsymbol{\mu}_p, \boldsymbol{\Sigma}_p)) = \|\boldsymbol{\mu}_q - \boldsymbol{\mu}_p\|_2^2 + \text{Tr}\left(\boldsymbol{\Sigma}_q + \boldsymbol{\Sigma}_p - 2(\boldsymbol{\Sigma}_q^{\frac{1}{2}} \boldsymbol{\Sigma}_p \boldsymbol{\Sigma}_q^{\frac{1}{2}})^{\frac{1}{2}}\right). \tag{E2}$$

The distance should be $0$ if the distributions perfectly correspond.

– The *mean-squared distance from the generated samples to the nearest test sample* measures the closeness of the test data to the distribution from the generated data. This metric is dependent on the number of training samples, as the implicit representation of the distribution depends on the number of samples.

$$\overline{d}_{\text{gen}}(q, p) = \mathbb{E}_{\boldsymbol{x} \sim q} \min_{\boldsymbol{z}_0 \sim p} \|\boldsymbol{x} - \boldsymbol{z}_0\|_2^2. \tag{E3}$$

The distance should be $0$ if the distributions perfectly correspond.

– The *mean-squared distance from the test samples to the nearest generated sample* measures the closeness of the generated data to the attractor of the dynamical system, represented by the test data. This metric is independent of the number of generated samples, as there is always the same number of test samples.

$$\overline{d}_{\text{test}}(q, p) = \mathbb{E}_{\boldsymbol{z}_0 \sim p} \min_{\boldsymbol{x} \sim q} \|\boldsymbol{x} - \boldsymbol{z}_0\|_2^2. \tag{E4}$$

The distance should be $0$ if the distributions perfectly correspond.

– As measure for *rare events*, we use a peak-over/under-threshold metric. We determine the 1st $p_{0.01}$ and 99th $p_{0.99}$ percentile along the three dimensions in the testing data and measure then how often events below (1st) or above (99th) these percentiles occur in the generated samples.

$$\text{POT}(q, p) = \mathbb{E}_{\boldsymbol{x} \sim q}\left[P(\boldsymbol{x} \leq p_{0.01}) + P(\boldsymbol{x} \geq p_{0.99})\right]. \tag{E5}$$

If the extremes are correctly represented, the expected value should be $0.02$

## Appendix F: Additional results for the learned representation

To determine if the denoising neural network can extract features about the dynamical system, we can fit linear models from known polynomial features to the features of the neural network. First, we first extract with the denoising neural network features from states in the training dataset. Similarly to the surrogate modelling experiments, Sect. 4.2, we use six different
pseudo-time steps. Secondly, we linearly regress from known polynomial features ($x$, $y$, $z$, $xy$, $xz$) to the extracted features, giving us the input coefficients $\boldsymbol{\beta}_{\text{in}}$. Thirdly, we linearly regress from the extracted features to the derivatives estimated with the Lorenz 1963 equations, Eq. (19a)–Eq. (19c), which gives us the output coefficients $\boldsymbol{\beta}_{\text{out}}$. Fourthly, we multiply the input coefficients $\boldsymbol{\beta}_{\text{in}}$ with the output coefficients $\boldsymbol{\beta}_{\text{out}}$, resulting into a linear factor table $\boldsymbol{\beta}$.





**Table F1.** Linear factors between polynomial features and time derivative of Lorenz 1963 system extracted with (a) an untrained diffusion model and (b) a pretrained diffusion model. The factors are rounded to the second decimal number. Bold values correspond to factors of the ODE that defines Lorenz 1963 system, while red values indicate a wrong factor up to the second decimal number. Negative zero values, like $-0.00$, result out of rounding to two decimal numbers.

(a) Untrained diffusion model

| Poly | $\dot{x}$ | $\dot{y}$ | $\dot{z}$ |
|------|------|------|------|
| x | **−10.00** | **−12.93** | −0.02 |
| y | **10.00** | **10.00** | 0.04 |
| z | 0.00 | −0.00 | **−0.00** |
| xy | 0.00 | 0.00 | **0.00** |
| xz | 0.00 | **0.00** | 0.00 |

(b) Pre-trained diffusion model

| Poly | $\dot{x}$ | $\dot{y}$ | $\dot{z}$ |
|------|------|------|------|
| x | **−10.00** | **28.00** | 0.00 |
| y | **10.00** | **−1.00** | −0.00 |
| z | −0.00 | −0.00 | **−2.67** |
| xy | −0.00 | 0.00 | **1.00** |
| xz | 0.00 | **−1.00** | −0.00 |

The polynomial features and their linear factors are well-defined by the Lorenz 1963 equations, Eq. (19a)–Eq. (19c). Con-
725 sequently, if the feature extractor has learned some meaningful features about the dynamical system, the linear factor table
$\boldsymbol{\beta}$ should recover the original factors from the model. We compare two feature extractor, an untrained extractor, where the
weights are randomly initialized, and a feature extractor, pre-trained as denoising diffusion model, Table F1. The linear factors
estimated with the pre-trained diffusion model fit almost perfectly those from the original Lorenz equations. By contrast, a
random neural network is unable to extract such features. This indicates that pre-training denoising diffusion models for state
generation can learn useful features about the dynamical system itself.

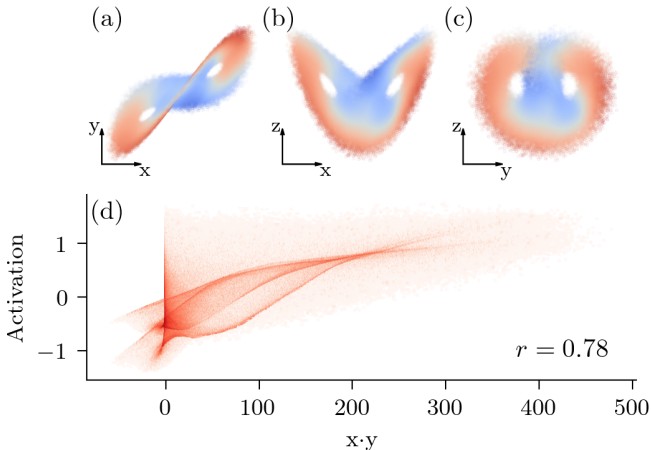

**Figure F1.** Visualization of a learned feature of a pre-trained diffusion model with a $v$ parameterization and a cosine noise scheduling. Shown is the activation of the feature with the highest correlation to the $x \cdot y$ product (a) projected onto the $x$-$y$ plane, (b) projected onto the $x$-$z$ plane, (c) projected onto the $y$-$z$ plane, and (d) projected to the $x \cdot y$ product.



The combined linear coefficients $\beta$ use all extracted features of the neural network. In Fig. F1, we show the feature that has the highest correlation ($r = 0.78$) to the $x \cdot y$ product. We can see a non-linear dependency between the extracted feature and the product. Consequently, although the product is used to estimate the time derivative of $z$, there is no single feature that linearly represents this product. Therefore, the features that represent the dynamical system are entangled in the features as represented by the feature extractor.

*Code and data availability.* The source code for the experiments and the neural networks is publicly available under https://github.com/cerea-daml/ddm-attractor. With the source code, the Lorenz 1963 data needed for the experiments can be reproduced. Exemplary weights for one diffusion model are additionally contained in the repository. The authors will provide further access to the weights of the neural networks upon request.

*Author contributions.* TSF had the original research idea. TSF, CD, AF, and MB refined this idea. TSF found in discussions with LD the idea for ensemble generation. TSF, LD, CD, AF, and MB analysed the results. TSF wrote the manuscript with LD, CD, AF, and MB reviewing.

*Competing interests.* The authors declare no competing interests.

*Acknowledgements.* The authors received financal support from the project GenD$^2$M funded by the LEFE-MANU program of the INSU/C-NRS and from the project SASIP (grant no. 353) funded by Schmidt Futures – a philanthropic initiative that seeks to improve societal
outcomes through the development of emerging science and technologies.



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
