# Peer review of "Representation learning with unconditional denoising diffusion models for dynamical systems"

_EGUsphere, 2023_

## Author Comment (AC1)

**Response to Referee 1**
**for "Representation learning with unconditional denoising diffusion models for dynamical systems"**

Finn, T. S., Disson, L., Farchi, A., Bocquet, M., and Durand, C.

24th May 2024

**RC: Reviewer Comment**, AR: Author Response

**RC:** **This research paper presents a study on using denoising diffusion models for data-driven representation learning of dynamical systems. The research demonstrates the utility of such networks with the Lorenz 63 system, showing that the trained network can produce samples almost indistinguishable from those on the attractor, indicating the network has learned an internal representation of the system. This representation is then used for surrogate modeling and generating ensembles out of a deterministic run. Overall I found this paper very well written and the contribution of introducing diffusion model into dynamical systems in geoscience novel and of clear contribution. Here lists my comments before I can recommend acceptance of this manuscript:**

**AR:** We thank Dr. Cheng for the constructive feedback on our manuscript, especially with the remarks related to small dimensionality of the here-tested Lorenz 1963 system. In the following, we will discuss the raised comments and indicate what we will change in the revised manuscript.

**RC:** **If I understand correctly, the objective of this study is to explore the possibility of using diffusion model for high-dimension systems in geoscience. The numerical experiments are carried out using a three dimensional Lorenz model. To enhance the discussion, It would be beneficial if the authors could explain how generalizable their approach is to a high-dimensional spatial temporal system (e.g. by adding CNN or transformer layers for feature extractions (encoding) and decoding etc).**

**AR:** The goal of this study is foremost to give a proof-of-concept on representation learning for dynamical systems with denoising diffusion models. We tackle the question,

what would happen if we would have much more data and much more parameters in denoising diffusion models than in the system. Consequently, we selected the Lorenz 1963 as system of interest as the system has only three dimensions and we can easily generate millions of data points and train networks with millions of parameters. Our results indicate that in such settings, denoising diffusion models can generalize to the system and be used for downstream tasks. While there is still the question if the generalization also holds for high-dimensional and large-scale systems, the results gives us hope that it can be the case. To take this comment into account, we will strengthen in the abstract and introduction the proof-of-concept character of the study. In Sect. 5 (Summary and Discussion), we will additionally elucidate more on an outlook how these might hold for higher-dimensional systems, e.g., hypothesizing what happens if we would add transformer layers.

RC: **As a consequence of the small dimension, the "latent space" in your diffusion model (256) is much larger the one of the physics space (3). Therefore, you have little risk in losing any information when using the denoising network for surrogate modelling. The authors may consider adding a baseline of transfer learning from an untrained (randomly initialized denoising NN) in Fig 7. The authors have shown the results of untrained NN in Tab 3 but only with a linear fine-tuning. What happens if you fine-tune with a non-linear NN of an untrained denoising NN?**

AR: The dimensionality of the feature space (avoiding latent space to circumvent issues with the latent/noised space from diffusion models) spanned by the denoising diffusion model is indeed much larger than the dimensionality of the system, a consequence of the study's character as proof-of-concept. Independently, the question that we answer is if this features space can be used for surrogate modelling. The trained diffusion model has the "right" features for surrogate modelling, whereas a randomly initialized model fails to have them. The correct features for surrogate modelling are hence learned and not by chance. Consequently, features that are needed to generate states on the system's attractor seem to be useful for surrogate modelling. We deliberately neglected the baseline of the untrained diffusion model in Fig. 7: the surrogate model with the untrained feature extractor rapidly converges to a nRMSE of 1 as also visible in Table 3. Including this baseline would not provide additional information to the table and could distract from the main message of the Figure that the trained features are more stable than random Fourier features. Consequently, in the revised version of manuscript, we will still omit this baseline from the Figure. For completeness, we nevertheless include the baseline in the modified Fig. 1 of this answer. As the linear probing already shows, the features extracted by the untrained diffusion model are unaligned to the dynamical system, hence, we neglected the experiment with the small NN. In the revised version of the manuscript, we will include the scores for this experiment.

RC: **In figure 7, it seems that the dense neural network with two layers trained**

> **from scratch outperforms your transfer learning from the diffusion model. Is that the case? In fact, results in tab 3 also show that the model trained from scratch (dense \*3 and resnet) performs similarly to the fine-tuning from your diffusion model? The authors may want to add some comments regarding this**

AR: Figure 7 shows the performance over long lead times and used here to show the stability of the surrogate models. Since the models were trained for lead times of $0.1\,\mathrm{MTU}$, we cannot expect that they perform as well for very long lead times. To improve the performance therein, one could apply autoregressive training steps as often done in surrogate models for the atmosphere, e.g. in GraphCast. Furthermore, the difference between the transfer learned surrogate model and the surrogate models learned from scratch are in fact smaller than the difference caused by different random seeds and might be a result from chance. Consequently, we stay at our claim that transfer learning can perform better than NNs trained from scratch. To nevertheless take the comment into account, we will add something like *"Since the models are trained for lead times of* $0.1\,\mathrm{MTU}$ *without autoregressive steps, their performance for longer lead times is heavily impacted by randomness as shown by the spread between seeds in Fig. 7. The difference between the NN models is much smaller than the effect of randomness, which makes it difficult to discriminate if the differences are by chance"* to the explanation of Fig. 7 in the manuscript.

RC: **Page 3, 'generative training is rarely used for pre-training and representation learning of high-dimensional systems'. There are some works tried to use diffusion model for contrastive models, e.g,**

— **Yang, X. and Wang, X., 2023. Diffusion model as representation learner. In Proceedings of the IEEE/CVF International Conference on Computer Vision (pp. 18938-18949).**

— **Mittal, S., Abstreiter, K., Bauer, S., Schölkopf, B. and Mehrjou, A., 2023, July. Diffusion based representation learning. In International Conference on Machine Learning (pp. 24963-24982). PMLR.**

**The authors may want to include some references and discuss the difference/similarity compared to the method used in this paper. This paper is probably the first one to propose diffusion-based representation learning in dynamical systems(?)**

AR: The intention of this specific sentence was to show the gap. We understand that this sentence might be missleading and will change it to "Since training deep generative models remains difficult yet, generative training is less often used for pre-training and representation learning of high-dimensional systems than other methods like contrastive learning (e.g., SimCLR from Chen et al., 2020)." A smaller literature review is given in the paragraph afterwards. Caused by the timeliness of the topic, we have however missed these recent publications and we will add them to the literature review, thanks for pointing to them. Hence, we will add: "Concurrently

to our study, Mittal el al., 2023 and Yang and Wang, 2023, propose to directly use denoising difufsion models for representation learning from images. However, to our knowledge, we are the first introducing these models for representation learning from dynamical systems."

**RC:** Page 9, 'show that this representation is entangled' why it is important for the learned features to be entangled?

AR: Entangled features are more difficult to interpret and also more difficult to use in downstream tasks, as indicated by the need of features from several pseudo times steps and a small NN for surrogate modelling. Consequently, this can be seen as one drawback of the learned representation.

**RC:** Page 11, check the sentence 'As we will see later, the bigger the Because of the state-dependency, the resulting distribution is implicitly represented by the ensemble and could extend beyond a Gaussian assumption'

AR: Thank you for spotting this incomplete sentence and left over from the internal revision process. We will remove the part "As we will see later, the bigger the " since it is covered in the next paragraph.

**RC:** Page 13, it seems that you have used a lot of training samples (1.6*E7) for your diffusion model for the Lorenz system of dimension 3. I was wondering if a standard surrogate model will require that much. That is saying maybe a standard surrogate model can outperform the diffusion-based one with less training data. I am curious to see the authors' thought.

AR: We used this many samples to be unconstrained from the training dataset size. It is very likely that much less training samples are needed for surrogate modelling and representation learning, yet, we do not know when forecast performance drops. While many samples might be needed to learn a representation, we agree with you that standard surrogate models need much less samples as they are more specialized. The premise of representation learning is however that the learned features can be then transferred to other problems like surrogate modelling, where we would need much less training samples than for representation learning, and possibly even less than for standard surrogate models. Since the forecast error of surrogate models in the Lorenz 1963 system is very low, this hypothesis should be tested with higher-dimensional and more difficult systems.

**RC:** fig 5 (a) and 1(b). if I understand correctly, the x-axis is the pseudo time instead of the real time in the dynamical system. if it is the case, it would be benificial to add an x-axis label to avoid any confusion.

AR: Yes the axis is in pseudo-time and we will add this labeling to avoid confusion, thanks for spotting this.

[Figure]

Figure 1: The normalized root-mean-squared error (nRMSE) as function of integration time steps for random Fourier features (RFF) with 1536 features and a linear regression, a *dense* neural network with two layers trained from scratch, and transfer learned models (Transfer) with features from six tipseudo-time steps with a linear regression and from two pseudo-time steps with a neural network. Shown is the median across ten different random seeds. Additionally, for the RFF (1536, linear) and the Transfer $(2 \times \tau, \text{NN})$ experiments, the 5th and 95th percentile is depicted as shading.

---

## Author Response (AR1)

**Author's response**
**for "Representation learning with unconditional denoising diffusion models for dynamical systems"**

Finn, T. S., Disson, L., Farchi, A., Bocquet, M., and Durand, C.

20th June 2024

**RC: Reviewer Comment**, AR: Author Response

**1 Response to Referee 1**

**RC:** **This research paper presents a study on using denoising diffusion models for data-driven representation learning of dynamical systems. The research demonstrates the utility of such networks with the Lorenz 63 system, showing that the trained network can produce samples almost indistinguishable from those on the attractor, indicating the network has learned an internal representation of the system. This representation is then used for surrogate modeling and generating ensembles out of a deterministic run. Overall I found this paper very well written and the contribution of introducing diffusion model into dynamical systems in geoscience novel and of clear contribution. Here lists my comments before I can recommend acceptance of this manuscript:**

AR: We thank Dr. Cheng for the constructive feedback on our manuscript, especially with the remarks related to small dimensionality of the here-tested Lorenz 1963 system. In the following, we will discuss the raised comments and indicate what we have changed in the revised manuscript.

**RC:** **If I understand correctly, the objective of this study is to explore the possibility of using diffusion model for high-dimension systems in geoscience. The numerical experiments are carried out using a three dimensional Lorenz model. To enhance the discussion, It would be beneficial if the authors could explain how generalizable their approach is to a high-dimensional spatial temporal system (e.g. by adding CNN or transformer layers for feature extractions (encoding) and decoding etc).**

AR: The goal of this study is foremost to give a proof-of-concept on representation learning for dynamical systems with denoising diffusion models. We tackle the question, what would happen if we would have much more data and much more parameters in denoising diffusion models than in the system. Consequently, we selected the Lorenz 1963 as system of interest as the system has only three dimensions and we can easily generate millions of data points and train networks with millions of parameters. Our results indicate that in such settings, denoising diffusion models can generalize to the system and be used for downstream tasks. While there is still the question if the generalization also holds for high-dimensional and large-scale systems, the results gives us hope that it can be the case. To take this comment into account, we have strengthen the proof-of-concept character of the study by adding the keyword *proof-of-concept* to the abstract and signifying that the study is done with *the Lorenz 1963 system* to the first paragraph of the introduction. We have added a paragraph in Sect. 5 (Summary and Discussion), elucidating more on how the proposed methods might perform for higher-dimensional systems.

**RC:** **As a consequence of the small dimension, the "latent space" in your diffusion model (256) is much larger the one of the physics space (3). Therefore, you have little risk in losing any information when using the denoising network for surrogate modelling. The authors may consider adding a baseline of transfer learning from an untrained (randomly initialized denoising NN) in Fig 7. The authors have shown the results of untrained NN in Tab 3 but only with a linear fine-tuning. What happens if you fine-tune with a non-linear NN of an untrained denoising NN?**

AR: The dimensionality of the feature space (avoiding latent space to circumvent issues with the latent/noised space from diffusion models) spanned by the denoising diffusion model is indeed much larger than the dimensionality of the system, a consequence of the study's character as proof-of-concept. Independently, the question that we answer is if this features space can be used for surrogate modelling. The trained diffusion model has the "right" features for surrogate modelling, whereas a randomly initialized model fails to have them. The correct features for surrogate modelling are hence learned and not by chance. Consequently, features that are needed to generate states on the system's attractor seem to be useful for surrogate modelling. We deliberately neglected the baseline of the untrained diffusion model in Fig. 7: the surrogate model with the untrained feature extractor rapidly converges to a nRMSE of 1 as also visible in Table 3. Including this baseline would not provide additional information to the table and could distract from the main message of the Figure that the trained features are more stable than random Fourier features. Consequently, in the revised version of manuscript, we have still omitted this baseline from the Figure. For completeness, we nevertheless include the baseline in the modified Fig. 1 of this answer. Additionally, we have added to Tab. 3 of the revised manuscript an experiment where we use a shallow neural

network after an untrained feature extractor.

**RC:** **In figure 7, it seems that the dense neural network with two layers trained from scratch outperforms your transfer learning from the diffusion model. Is that the case? In fact, results in tab 3 also show that the model trained from scratch (dense \*3 and resnet) performs similarly to the fine-tuning from your diffusion model? The authors may want to add some comments regarding this**

AR: Figure 7 shows the performance over long lead times and used here to show the stability of the surrogate models. Since the models were trained for lead times of 0.1 MTU, we cannot expect that they perform as well for very long lead times. To improve the performance therein, one could apply autoregressive training steps as often done in surrogate models for the atmosphere, e.g. in GraphCast. Furthermore, the difference between the transfer learned surrogate model and the surrogate models learned from scratch are in fact smaller than the difference caused by different random seeds and might be a result from chance. Consequently, we stay at our claim that transfer learning can perform similarly or better than NNs trained from scratch. To nevertheless account for the comment, we have revised the result section for the surrogate models and added *"Since the models are trained for lead times of 0.1 MTU without autoregressive steps, their performance for longer lead times is impacted by randomness as shown by the spread between difference seeds in Fig. 7. Compared to this spread, the transfer learned models and the NNs trained from scratch perform similarly"*.

**RC:** **Page 3, 'generative training is rarely used for pre-training and representation learning of high-dimensional systems'. There are some works tried to use diffusion model for contrastive models, e.g,**

  – **Yang, X. and Wang, X., 2023. Diffusion model as representation learner. In Proceedings of the IEEE/CVF International Conference on Computer Vision (pp. 18938-18949).**

  – **Mittal, S., Abstreiter, K., Bauer, S., Schölkopf, B. and Mehrjou, A., 2023, July. Diffusion based representation learning. In International Conference on Machine Learning (pp. 24963-24982). PMLR.**

**The authors may want to include some references and discuss the difference/similarity compared to the method used in this paper. This paper is probably the first one to propose diffusion-based representation learning in dynamical systems(?)**

AR: The intention of this specific sentence was to show the gap. We understand that this sentence might be misleading and have change it to *"generative training is rarely less often used for pre-training and representation learning of high-dimensional systems than other methods like contrastive learning (e.g., Chen et al., 2020)"*. A smaller literature review is given in the paragraph afterwards. Caused by the

timeliness of the topic, we have however missed these recent publications and we have added them to the literature review, thanks for pointing to them. Hence, we have added to the introduction: *"Concurrently to this study, Mittal et al. (2023); Yang and Wang (2023) propose to directly use DDMs for representation learning from images. However, to our knowledge, we are the first introducing these models for representation learning from dynamical systems"*.

**RC:** **Page 9, 'show that this representation is entangled' why it is important for the learned features to be entangled?**

AR: Entangled features are more difficult to interpret and also more difficult to use in downstream tasks, as indicated by the need of features from several pseudo times steps and a small NN for surrogate modelling. Consequently, this can be seen as one drawback of the learned representation.

**RC:** **Page 11, check the sentence 'As we will see later, the bigger the Because of the state-dependency, the resulting distribution is implicitly represented by the ensemble and could extend beyond a Gaussian assumption'**

AR: Thank you for spotting this incomplete sentence and left over from the internal revision process. We have removed the part *"As we will see later, the bigger the"* since it is covered in the next paragraph.

**RC:** **Page 13, it seems that you have used a lot of training samples (1.6*E7) for your diffusion model for the Lorenz system of dimension 3. I was wondering if a standard surrogate model will require that much. That is saying maybe a standard surrogate model can outperform the diffusion-based one with less training data. I am curious to see the authors' thought.**

AR: We used this many samples to be unconstrained from the training dataset size. It is very likely that much less training samples are needed for surrogate modelling and representation learning, yet, we do not know when forecast performance drops. While many samples might be needed to learn a representation, we agree with you that standard surrogate models need much less samples as they are more specialized. The premise of representation learning is however that the learned features can be then transferred to other problems like surrogate modelling, where we would need much less training samples than for representation learning, and possibly even less than for standard surrogate models. Since the forecast error of surrogate models in the Lorenz 1963 system is very low, this hypothesis should be tested with higher-dimensional and more difficult systems.

**RC:** **fig 5 (a) and 1(b). if I understand correctly, the x-axis is the pseudo time instead of the real time in the dynamical system. if it is the case, it would be benificial to add an x-axis label to avoid any confusion.**

AR: Yes the axis is in pseudo-time and we have added this labeling in Fig. 1 and Fig. 5 to avoid confusion, thanks for spotting this.

**2 Response to Referee 2**

RC: **This is a very interesting and novel study on the use of denoising diffusion model for representation learning. The manuscript is well written and describes very nicely the context, how these approaches (rooted in image applications) can be adapted to geosciences, and illustrates two distinct relevant applications, surrogate modelling and ensemble generations, that are both extremely important in high dimensional settings.**
**I think the manuscript can be accepted almost as it is, but I have a few minor comments I would encourage the Authors to look at.**

AR: We thank the second referee for the constructive feedback on our manuscript, especially with the remarks related to the theory of dynamical systems. In the following, we discuss the raised comments and indicate what we have changed in the revised manuscript.

RC: **While there are little spaces for doubts, I would strongly suggest the Authors to specify that their approach applies to ergodic chaotic dynamics for which an invariant distribution exists that describe the state distribution on the system's attractor. An obvious counterexample would be a stable system having an equilibrium point (or a limit cycle) as attractor.**

AR: Thank you for the suggestion to sharpen the focus of the manuscript around ergodic chaotic dynamics. To sharpen the introduction and take this suggestion into account, we have added to the second paragraph in the introduction *"This generation process is expected to be successful if there is an invariant state distribution on the system's attractor, which exists for ergodic chaotic dynamics"*

RC: **When mentioning the Schrodinger Bridge (page 2), you may want to refer to Reich S. 2019 (doi:10.1017/S0962492919000011) as an exemplar study of the same analogy but in the area of data assimilation.**

AR: Thank you for pointing us to the seminal work of S. Reich on the connection between the Schrödinger Bridge and data assimilation. Originally, we left this work out as our study is focused on machine learning. However, since we apply diffusion models on a data assimilation problem, we have added this work to the third paragraph in the introduction by *"Notably, the Schrödinger Bridge has been already exploited for discrete- and continuous-time data assimilation (Reich, 2019)"*.

**RC:  Line 27. "..dynamical systemS ..."**

AR: Thank you for spotting this inconsistency, we have used the plural of dynamical systems as proposed.

**RC:  In the caption of Fig1b, use (left/right) to point the reader.**

AR: Thank you for the suggestion, we have added (left) and (right) to the caption of Fig. 1.

**RC:  Line 44. I think you should always order references chronologically.**

AR: Thank you for spotting the chronological inconsistency here, we have swapped the citations.

**RC:  Line 53–59. While I understand and I like the Authors narrative and choice of references. Nevertheless, and particularly for the readers of NPG, it would be appropriate to also mention the large bulk of work on the generation of ensemble members based on dynamical systems's theory and data assimilation. A good recent reference is 10.1029/2021MS002828**

AR: Thank you for the suggestion of adding references about the generation of ensemble members based on the properties of the dynamical system. To take these works into account, we have added to the paragraph *"Another method to generate an ensemble for data assimilation would be to make use of the knowledge about the system's error propagation, in form of singular vectors (Molteni et al., 1996) or bred vectors (Toth and Kalnay, 1993), as similarly used to initialize ensemble weather forecasts (Buizza et al., 2005) or sub-seasonal forecasts (Demaeyer et al., 2022)"*.

**RC:  I am a bit of an inconvenience with the use of the term "latent". On the one side I agree with a comment from the other Reviewer. On the other I do also see in line 100 that you state z=x which makes one deduce the latent and actual state have the same dimension. Finally, while it is true that latent variables are defined in relation to their indirect (often hidden) relation with the observables quantities, with no reference to their number (or space dimension), in many practical applications the latent space is assumed/defined/used as being of smaller dimension.**

AR: We agree with both referees about the confusion with the latent space of diffusion models, especially if they are used for representation learning where the features of the trained NN span another space. The latent space of diffusion models usually describes the space of noised data, while the other is rather a *feature space*. The noised data space is called latent space as diffusion models can be seen as hierarchical variational autoencoder (Luo, 2022), where one can also learn a mapping from data space into this latent space with a possibly reduced dimensionality (e.g., Rombach et al., 2022). Diffusion models that act in data space can be seen as special version of these variational diffusion models, where the mapping between spaces is the identity function.

**RC: Line 115. I would add ".... prior distribution FOR THE DENOISING PROCESS."**

AR: Thank you for the suggestion and we have added *"for the diffusion process"* to make the writing clearer.

**RC: Equations (8). Wouldn't be better to (re)state clearly that we do not have access to x in practice?**

AR: Thank you for the suggestion and we have added *"Note, during generation, the state* $\mathbf{x}$ *is unknown, and we have to approximate Eq. (8b) to generate data, as we discuss in the following"* to clarify the use of Eq. 8 for generation.

**RC: Line 145. Is that because they do not depend on x?**

AR: As the neural network just approximates the denoised state by Tweedie's formula, we have defined in Eq. (10), that the covariances between the analytical denoising step and the approximated step match, other definition are possible though (e.g., Ho et al., 2020, Nichol and Dhariwal, 2021). With the definition of the diffusion process, we can then specify the Kullback-Leibler divergence with different quantities, e.g., if we would directly predict the state $\mathbf{x}_\theta(\mathbf{z}_\tau, \tau)$ or the noise $\boldsymbol{\epsilon}_\theta(\mathbf{z}_\tau, \tau)$. The simplification of the loss can be traced in Ho et al. (2020) or in Luo (2022).

**RC: Line 153. I think "Equation" must be written at the beginning of the sentence.**

AR: Thank you for spotting this, yes, we agree that *Equation* should be written at the beginning of a sentence and we have corrected the sentence accordingly.

**RC: Line 176. Instead of "normally" I would suggest "most of the times".**

AR: Thank you for the nice suggestion, *"most of the times"* sounds indeed better and we have changed the words accordingly.

**3 General changes in the manuscript**

While incorporating the additional experiment into Tab. 3, we decided to revise the ordering and structure of Section 4.2.1, exceeding the comment from Referee 1. The changes in this subsection have no impact on the results and their conclusions.

[Figure]

Figure 1: The normalized root-mean-squared error (nRMSE) as function of integration time steps for random Fourier features (RFF) with 1536 features and a linear regression, a *dense* neural network with two layers trained from scratch, and transfer learned models (Transfer) with features from six tipseudo-time steps with a linear regression and from two pseudo-time steps with a neural network. Shown is the median across ten different random seeds. Additionally, for the RFF (1536, linear) and the Transfer ($2 \times \tau$, NN) experiments, the 5th and 95th percentile is depicted as shading.